# `metaTextGrad`: Automatically optimizing language model optimizers

**Guowei Xu**
Tsinghua University

**Mert Yuksekgonul**
Stanford University

**Carlos Guestrin**
Stanford University

**James Zou**
Stanford University

## Abstract

Large language models (LLMs) are increasingly used in learning algorithms, evaluations, and optimization tasks. Recent studies have shown that using LLM-based optimizers to automatically optimize model prompts, demonstrations, predictions themselves, or other components can significantly enhance the performance of AI systems, as demonstrated by frameworks such as DSPy and TextGrad. However, optimizers built on language models themselves are usually designed by humans with manual design choices; optimizers themselves are not optimized. Moreover, these optimizers are general purpose by design, to be useful to a broad audience, and are not tailored for specific tasks. To address these challenges, we propose `metaTextGrad`, which focuses on designing a meta-optimizer to further enhance existing optimizers and align them to be good optimizers for a given task. Our approach consists of two key components: a meta prompt optimizer and a meta structure optimizer. The combination of these two significantly improves performance across multiple benchmarks, achieving an average absolute performance improvement of up to 6% compared to the best baseline.

## 1 Introduction

Large language models (LLMs) are increasingly used in learning algorithms, optimization, and evaluation tasks [1, 2, 3, 4, 5]. However, algorithms that incorporate LLMs often face significant challenges. Firstly, many of these algorithms are still hand-crafted to a considerable extent, requiring substantial human expertise and effort to design and implement effectively. Second, LLMs are notably sensitive to the specific wording and structure of their instructions [6], making it tedious to improve them effectively to be used in learning algorithms.

Many studies have explored prompt optimization approaches to automatically design better prompts and enhance the performance of LLMs. For example, algorithms such as OPRO [3], MIPRO [7], and TextGrad [1] have introduced optimizers based on LLMs that support automatic prompt optimization. However, these optimizers are often fixed, applying the same optimization strategies with the same optimizer prompts independent of the task, lacking a process to align with the specific task. Importantly, these optimizers are general purpose by design, to be useful to many different downstream tasks and be used by large user bases. Furthermore, different optimizers likely excel at different types of optimization tasks; thus, achieving the best of all optimizers introduces either a choice to be made among them or a strategy to ensemble them. There is no method to automatically design and improve the optimizers to call based on the characteristics of a task.

In this work, we explore how to automatically optimize both the structure and the prompt of optimizers. Specifically, we assume that all LLM calls are black-box calls, where the internal states of the LLM,

39th Conference on Neural Information Processing Systems (NeurIPS 2025).

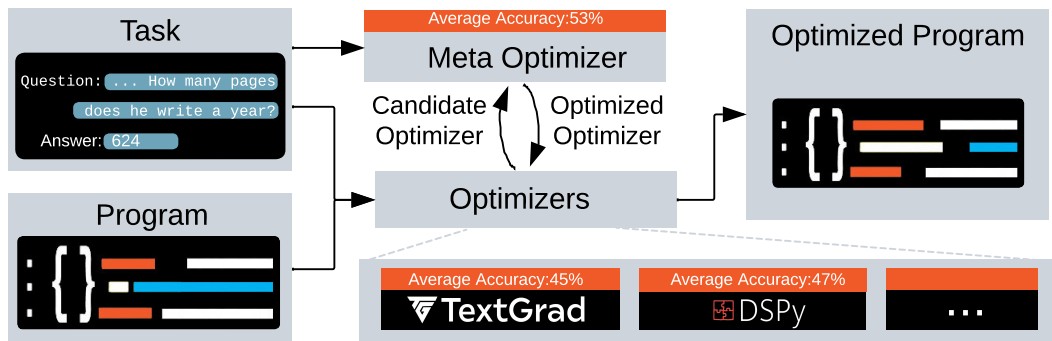

Figure 1: Illustration of the meta-optimization process. A meta-optimizer optimizes LLM optimizers by aligning them with specific tasks through task interaction while leveraging the strengths of different optimizers to propose a more effective optimizer.

as well as any information such as model gradients, are inaccessible, and only the inputs to and outputs of the LLM are observable. For the task to be optimized, we require only a small training dataset containing some input-output pairs and an evaluation metric the user wants to improve upon.

We propose using a meta-optimizer to automatically optimize the optimizer, with the objective of identifying improved optimizers such that the program optimized by it achieves the best performance on the given task. To this end, we introduce two types of meta-optimization strategies: the meta prompt optimizer and the meta structure optimizer. The meta prompt optimizer focuses on refining the prompts of the LLM optimizers to enhance their effectiveness and better align them with specific tasks. Meanwhile, the meta structure optimizer is designed to automatically determine the optimal combination and sequence of different optimizers or modules based on the characteristics of the task.

By combining these two types of meta-optimizers, we propose `metaTextGrad`, which integrates both components into a unified framework. Specifically, given a set of input optimizers, `metaTextGrad` first performs prompt optimization for each optimizer independently. Then, it explores the combination and sequencing of these optimizers to construct a more effective composite optimizer. We conducted experiments on multiple benchmarks, and the results demonstrate that our meta-optimized optimizers consistently outperform the existing ones.

Overall, we summarize our contributions as follows. First, we introduce the concept of meta-optimization, highlighting that existing LLM optimizers often require further task alignment and effective combination through a meta-optimizer to maximize their potential. Second, we develop two types of meta-optimizers: the meta prompt optimizer and the meta structure optimizer. Building on these, we propose `metaTextGrad`, which integrates both types of meta-optimizers into a unified framework. Lastly, experimental results on multiple benchmarks demonstrate that our method significantly outperforms baseline approaches in both performance and generalization.

## 2  Problem Statement

Here, we will follow the notation from [7]. Consider an LLM program $\Phi$ that may contain multiple LLM calls forming a pipeline, with each call using a different prompt. The pipeline structure of a program and the prompt corresponding to each LLM call can be learnable and optimized by an LLM optimizer.

The task of the LLM optimizer is to find the optimal program $\Phi$ given a training dataset $\mathcal{D}$ (with pairs of inputs $x$ and outputs $y$), and an evaluation metric $\mu$.

$$\Phi^* = \arg\max_{\Phi} \frac{1}{|\mathcal{D}|} \sum_{(x,y)\in\mathcal{D}} \mu(\Phi(x), y) \tag{1}$$

Existing methods attempt to approximate solutions to this optimization problem from different perspectives. Optimizers such as MIPRO [7], OPRO [8], and TGD [9] aim to find better prompts, while optimizers like ADAS [10] focus on exploring structures. We unify the existing optimizer

algorithms into a general framework, as outlined in Algorithm 1. Importantly, these optimizers are the same type as of $M$, are designed by humans, and their structure and prompts are fixed.

Specifically, each optimizer should be capable of performing four operations:

1. **Initialize**: Given the training dataset and the program to be optimized, prepare the training scheme.

2. **Propose**: Suggest improvements to the program, which may involve modifications to the pipeline structure, prompts, or other aspects.

3.**Update**: Update the current optimal program based on the evaluation results of the improved program and determine the next proposal.

4. **ExtractOptimizedProgram**: Return the best program found so far.

---

**Algorithm 1** Inner Loop: Optimize $\Phi$ with Optimizer $M$

---

1: **Input:** Optimizer $M$, Initial Program $\Phi$, Max Iterations $I$
2: **Input:** Training Data $\mathcal{D}$, Validation Data $\mathcal{D}_{val}$, Metric $\mu$
3: **Output:** $\Phi^*$, i.e., the optimized version of $\Phi$
4: $M$.Initialize$(\mathcal{D}, \Phi)$
5: **for** $k \leftarrow 1$ **to** $I$ **do**
6: $\quad \Phi_k \leftarrow M$.Propose()
7: $\quad \sigma \leftarrow \frac{1}{|\mathcal{D}_{val}|} \sum_{(x,y) \in \mathcal{D}_{val}} \mu\big(\Phi_k(x), y\big)$
8: $\quad M$.Update$(\Phi_k, \sigma)$
9: **end for**
10: $(\Phi^*, \sigma^*) \leftarrow M$.ExtractOptimizedProgram()
11: **return** $\big(\Phi^*, \sigma^*\big)$

---

The problem we investigate is how to further enhance existing optimizers and align them to be effective optimizers for a given task, instead of relying on human-written optimizers. To this end, we introduce the concept of a *meta-optimizer*. Let the optimized program obtained by an optimizer $M$ be denoted as $\Phi^* = M$.optimize$(\mathcal{D}, \Phi)$. The optimization objective of the meta-optimizer is to find improved optimizers such that the program optimized by this optimizer performs better on the corresponding task. In particular, the optimization problem is formalized as:

$$M^* = \arg\max_M \frac{1}{|\mathcal{D}|} \sum_{(x,y) \in \mathcal{D}} \mu(M.\text{optimize}(\mathcal{D}, \Phi)(x), y). \tag{2}$$

In optimization, good initializations often contribute to improved performance, in continuous or discrete optimization alike [11, 12]. Ideally, the meta-optimization framework should leverage the existing, manually designed optimizers and achieve further improvements. Therefore, the input to the meta-optimizer can include one or more existing optimizers as initialization, ultimately producing an optimized optimizer. The detailed process is illustrated in Algorithm 2.

---

**Algorithm 2** Meta-Optimization of Optimizers

---

1: **Input:** Meta-Optimizer $\widehat{M}$
2: **Input:** Max Meta-Iterations $J$, Max Inner-Iterations $I$
3: **Input:** Initial optimizers $\{M^{(1)}, M^{(2)}, \ldots, M^{(r)}\}$
4: **Input:** Training Data $\mathcal{D}$, Validation Data $\mathcal{D}_{val}$
5: **Input:** Metric $\mu$, Initial Program $\Phi$
6: **Output:** Optimized optimizer $M^*$
7: $\widehat{M}$.Initialize$(\mathcal{D}, \{M^{(i)}\}_{i=1}^r)$
8: **for** $j \leftarrow 1$ **to** $J$ **do**
9: $\quad M_j \leftarrow \widehat{M}$.Propose()
10: $\quad (\Phi_j^*, \sigma_j) \leftarrow$ InnerLoop$\big(M_j, \Phi, I, \mathcal{D}, \mathcal{D}_{val}, \mu\big)$
11: $\quad \widehat{M}$.Update$\big(M_j, \sigma_j\big)$
12: **end for**
13: $M^* \leftarrow \widehat{M}$.ExtractOptimizedOptimizer()
14: **return** $M^*$

---

Specifically, the meta-optimizer combines different input optimizers to propose new optimizers. The newly proposed optimizers execute the process described in Algorithm 1 to obtain the *inner loop*

optimization results. Based on the performance of the newly designed optimizers, the meta-optimizer updates the current best optimizer and determines the proposal for the next iteration (*outer loop*).

However, similar to the problem of optimizing a program, finding an optimal optimizer using a meta-optimizer is generally an intractable optimization problem. Therefore, it is necessary to introduce appropriate parameterizations and simplifications to the problem. In our work, we primarily focus on two types of parameterizations for the meta-optimizer:

1. automatically optimizing the prompts in the optimizer to align it with the given task (*Meta Prompt Optimizer*),

2. automatically optimizing the combination of different optimizers based on the characteristics of the task, forming a new composite optimizer (*Meta Structure Optimizer*).

Collectively, we can meta-optimize using both parameterizations to obtain better optimizers.

## 3 `metaTextGrad`: Automatically Optimizing Language Model Optimizers

In this section, we first introduce the theoretical insights behind the meta optimizer. Next, we further analyze and illustrate our motivation through concrete examples. Finally, we present the `metaTextGrad` pipeline, which consists of the meta prompt optimizer and the meta structure optimizer.

### 3.1 Theoretical Insight

We present the theoretical motivation for meta optimization, highlighting the importance of aligning the optimizer with the target task. In particular, it is shown that when both the training and test sets are sampled from the same underlying data distribution, an optimizer properly aligned via meta-learning on the training set will, with high probability, produce programs on the test set whose accuracy closely approaches that of the optimal optimizer. In contrast, an optimizer that has not been optimized on the training set lacks such theoretical guarantee.

We begin by introducing the necessary notation. Let $\mu : \mathcal{X} \to [0, 1]$ denote the accuracy metric, where $\mathcal{X}$ is the space of natural language outputs. Let $D$ be a distribution over natural language inputs $x$. Let $\Phi_\theta$ be an optimizer parameterized by $\theta \in \Theta$. Given input $x$ and access to $\mu$, $\Phi_\theta$ makes $T$ zeroth-order queries and returns an optimized output $x_T$.

Define the loss of the optimizer on input $x$ as $L(\theta, x) = 1 - \mu(x_T) = 1 - \mu(\Phi_\theta(x, T))$. The population loss is given by $R(\theta) = \mathbb{E}_{x \sim D}[L(\theta, x)]$. Let $S = \{x_1, \ldots, x_n\} \sim D^n$ denote a dataset sampled from $D$, and define the empirical loss on $S$ as $R_S(\theta) = \frac{1}{n} \sum_{i=1}^{n} L(\theta, x_i)$.

Let $\widehat{\theta}_S = \arg\min_{\theta \in \Theta} R_S(\theta)$ be the optimizer parameters obtained via meta-learning on the training set $S$, and let $\theta^* = \arg\min_{\theta \in \Theta} R(\theta)$ be the globally optimal optimizer under distribution $D$. We can now state the following theorem:

**Theorem 1.** *Let $S_1$ and $S_2$ be two datasets sampled independently from distribution $D$, with sizes $n$ and $m$, respectively. Then, with probability at least $1 - \delta$, the optimizer $\widehat{\theta}_{S_1}$ trained on $S_1$ satisfies:*

$$R_{S_2}(\widehat{\theta}) \leq R(\theta^*) + \sqrt{\frac{2\log(6/\delta)}{n}} + \sqrt{\frac{\log(6/\delta)}{2m}}. \tag{3}$$

The proof of Theorem 1 is based on Hoeffding's inequality, and the full derivation is provided in Appendix B. The theorem highlights the necessity of performing meta-optimization.

In contrast, an optimizer $\theta_0$ that hasn't been optimized on the training dataset has no theoretical guarantee. We can still guarantee that $R_{S_2}(\theta_0)$ is similar to $R(\theta_0)$, but the guarantee says nothing about how large $R(\theta_0)$ is. If the optimizer is bad on average for this task, it will still be bad on $S_2$ with high probability. This provides a theoretical foundation for the design of `metaTextGrad`.

### 3.2 Motivating Example

Existing optimizers such as TextGrad and DSPy are manually designed by humans with the goal of performing well across a broad distribution of tasks, and they indeed demonstrate strong average performance.

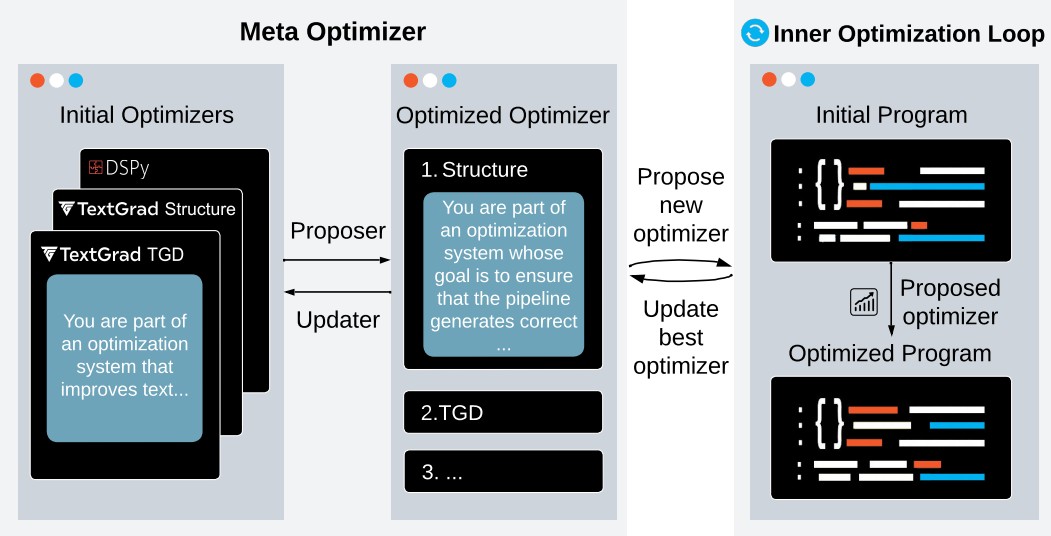

Figure 2: Illustration of `metaTextGrad`. `metaTextGrad` combines a meta prompt optimizer and a meta structure optimizer. Given a set of optimizers, `metaTextGrad` performs optimization in two steps. First, it individually refines each optimizer by optimizing its prompts to better align with the task. Then, it combines the different prompt-optimized optimizers to construct the final optimizer.

For example, the prompt used by the TextGrad TGD optimizer is as follows:

> **TextGrad TGD optimizer Prompt**
>
> You are part of an optimization system that improves text (i.e., variable). You will be asked to creatively and critically improve prompts, solutions to problems, code, or any other text-based variable.

As can be seen, this prompt is indeed highly general, with phrasing such as *'improve prompts, solutions to problems, code, or any other text-based variable'*.

However, in some cases, such general-purpose prompt fails to effectively optimize model performance. While the optimizers in both TextGrad and DSPy can obtain learning signals from task-specific evaluators, these signals tend to be noisy and sparse.

First, when feedback is provided solely in the form of scalar scores rather than textual guidance, it becomes sparse, making optimization substantially more challenging. Second, although optimizers such as TextGrad's TGD can accept textual gradient feedback, such feedback is often highly noisy, making it difficult for the optimizer to learn effectively. For example, the evaluator feedback received by the TGD optimizer may look like the following:

> **Feedback Received by the TGD optimizer**
>
> To improve the prompt for the executer and enhance the objective function, consider the following feedback: 1. **Explicit Criteria Definition**: The prompt should explicitly instruct the executer to define the criteria for optical activity at the beginning of the response. This can prevent ambiguity and ensure that the executer uses the correct scientific principles. For example, the prompt could include a directive to "List the criteria for optical activity before analyzing each compound." 2. **Data Verification Directive**: Incorporate a step in the prompt that requires the executer to verify the input data against reliable sources. This could be phrased as "Cross-check the properties of each compound with a trusted chemical database before proceeding with the analysis." 3. **Structured Logical Reasoning**: Encourage a structured approach to reasoning by breaking down the analysis into distinct steps. The prompt could suggest a format like "For each compound, first identify chiral centers, then assess symmetry, and finally determine optical activity".

> **Feedback Received by the TGD optimizer (continued)**
>
> 4. **Cross-Referencing Encouragement**: ... By incorporating these elements into the prompt, the executer can be guided to produce more accurate and reliable responses, thereby improving alignment with the ground truth answer and enhancing the objective function.

Here, the evaluator feedback includes suggestions such as *'List the criteria for optical activity before analyzing each compound'*, which are specific to a single problem instance. Such feedback is clearly noisy. Since a generic optimizer relies solely on feedback to guide its updates, it is likely to incorporate such suggestions into the optimized LLM program prompt, which can be detrimental to the overall performance of the program.

Yet, in reality, we can choose to let the LLM optimizer adapt to a specific task distribution in order to achieve better performance. This is because if the distribution of programs generated by the task-specific optimizer is more aligned with the task requirements, the difficulty of finding the optimal program will be significantly reduced, and the impact of noise in the evaluator's signal will be greatly mitigated, even when the feedback is noisy.

We illustrate this using the BBH Dyck Languages task as an example, showing that aligning the LLM optimizer to a specific task distribution can lead to improved performance. For instance, consider the following task-specific optimizer prompt:

> **A Task-specific Optimizer Prompt**
>
> You are part of an optimization system specialized in improving prompts for bracket matching and sequence completion tasks. Your role is to enhance prompts that help solve Dyck language problems, which involve proper nesting and closure of different types of brackets (, <>, ()). When improving prompts, focus on these critical aspects: (1) maintaining accurate bracket pair matching, (2) preserving the LIFO (Last In First Out) order of nested structures, (3) handling multiple bracket types simultaneously, and (4) ensuring complete closure of all open brackets. You should critically analyze how the prompts can better guide the model to track open brackets, maintain proper nesting order, and systematically complete sequences. Consider incorporating pattern recognition strategies and explicit validation rules in the improved prompts. Your improvements should lead to more reliable and accurate bracket sequence completions.

It can be observed that the task-specific optimizer leads to a shift in the distribution of LLM programs it tends to optimize, making it more likely to generate content that aligns with key task requirements, such as producing programs that satisfy the requirement of preserving the LIFO (Last In First Out) order, etc. As a result, even if the evaluator feedback is somewhat noisy or sparse, the LLM program optimized by the optimizer can still perform well, and is more likely to generate critical statements such as: Explicitly push each opening symbol onto the stack and pop it when a corresponding closing symbol is encountered. After processing each symbol, describe the current state of the stack, focusing on unmatched opening symbols. In contrast, if a generic optimizer is used, it becomes difficult to generate effective prompts under noisy or sparse feedback conditions.

This demonstrates that adapting to a new task distribution is meaningful. To avoid adapting to each new task distribution by hand, we meta-learn how to adapt, which is the core motivation of `metaTextGrad`.

### 3.3 Pipeline

#### 3.3.1 Meta Prompt Optimizer

LLM optimizers are usually designed by humans with manual design choices. As a result, the optimizers themselves are not optimized or aligned with a given task. We aim to optimize the prompts of LLM optimizers to further enhance their effectiveness and align them with tasks.

Below is a brief introduction to the implementation method of the meta prompt optimizer. The detailed pseudocode can be found in Appendix A.1. To **Initialize**, the meta prompt optimizer runs a round of optimization on the validation dataset to evaluate and record the initial performance of

the optimizer. To **Propose**, the meta prompt optimizer randomly samples data examples from the training dataset and analyzes the general characteristics of the task type. Based on the current best optimizer prompt, it proposes an improved prompt that is more aligned with the task. To **Update**, the optimized optimizer undergoes an inner loop optimization test on the validation dataset. If the test results outperform those of the previously best optimizer, the optimizer is updated. To **ExtractOptimizedOptimizer**, the meta prompt optimizer returns the best optimizer learned so far. Among these steps, **Propose** is the core of the meta prompt optimizer and the only stage where LLM calls are invoked. For detailed prompts, refer to Appendix D.1.

### 3.3.2 Meta Structure Optimizer

A variety of LLM optimizers have been proposed to optimize program structures, prompts, and other components. The meta structure optimizer is designed to automatically optimize the combination and ordering of different optimizers based on the characteristics of the task.

Below, we briefly introduce the working principles of the meta structure optimizer. The detailed pseudocode implementation can be found in Appendix A.2. To **Initialize**, the meta structure optimizer runs one round of optimization for each input optimizer on the validation dataset, selects the highest score and the corresponding optimizer as the initial best value. To **Propose**, we provide the meta structure optimizer with a set of reference optimizers. If previously optimized, better-performing optimizers exist, they are also included. Based on this, the meta structure optimizer integrates and proposes an improved optimizer. To **Update**, the meta structure optimizer evaluates whether the proposed optimizer shows improved performance on the validation dataset and updates the current best optimizer and score. To **ExtractOptimizedOptimizer**, the meta structure optimizer returns the best optimizer learned so far. The **Propose** stage is the only stage with LLM calls. For detailed prompts, please refer to Appendix D.2.

### 3.3.3 `metaTextGrad`

In the previous two sections, we introduced the meta prompt optimizer and the meta structure optimizer. `metaTextGrad` is composed of these two components. Specifically, after receiving a set of input optimizers, `metaTextGrad` first performs prompt optimization for each optimizer individually and then explores the combination of different optimizers to form a better composite optimizer.

## 4 Experiment

### 4.1 Experimental Setup

In this section, we use the existing optimizers from DSPy and TextGrad as baseline methods. We evaluate our approach and the baselines on multiple benchmarks, including BBH [13, 14], MMLU [15], and GPQA [16]. To ensure reproducibility, we provide the prompts and structures of the learned programs in Appendix E.

**Baselines.** We used zero-shot CoT, few-shot CoT [17], self-consistency [18], best-of-N [19, 20], MIPROv2 [7], TextGrad TGD [1] and ADAS-TG [10] as baselines. Zero-shot CoT relies on direct chain-of-thought reasoning, whereas MIPRO and TextGrad's TGD optimizer refine the program's prompt. ADAS is an algorithm for automatically searching and optimizing the program's structure. We implemented this algorithm within TextGrad as a baseline, referred to as ADAS-TG.

**Benchmarks.** We evaluate our method on four widely used, challenging, and diverse benchmarks: BBH Word Sorting, BBH Dyck Languages [13, 14], MMLU Abstract Algebra [15], and GPQA Diamond [16]. For detailed settings, please refer to Appendix C. Due to the non-determinism of LLM APIs [21], the test accuracy for each benchmark is averaged over five random seeds.

In our experiments, we consider three different levels of LLM calls: (1) LLM calls within the program itself, (2) LLM calls made by the optimizer while refining the program, and (3) LLM calls made by the meta-optimizer when optimizing the optimizer. As shown in Section 4.3, the frequency of these calls decreases significantly across these levels. This hierarchical structure allows for cost-effective resource allocation: the program should use a relatively economical model, the optimizer can leverage a more capable model, and the meta-optimizer should employ the best available model. Consequently,

| Method | Word Sorting | | Dyck Languages | | GPQA Diamond | | Abstract Algebra | | Average | |
|---|---|---|---|---|---|---|---|---|---|---|
| | Val | Test | Val | Test | Val | Test | Val | Test | Val | Test |
| **Vanilla prompting methods** | | | | | | | | | | |
| Zero-shot CoT | 0.46 | 0.55 | 0.06 | 0.05 | 0.32 | 0.34 | 0.74 | 0.70 | 0.40 | 0.41 |
| 8-shot CoT | 0.50 | 0.52 | 0.14 | 0.19 | 0.32 | 0.35 | 0.65 | 0.71 | 0.40 | 0.44 |
| Self-consistency (8) | 0.47 | 0.52 | 0.10 | 0.12 | 0.40 | **0.42** | 0.76 | 0.70 | 0.43 | 0.44 |
| Best of N (8) | 0.48 | 0.52 | 0.14 | 0.17 | 0.37 | 0.40 | 0.77 | 0.74 | 0.44 | 0.46 |
| **TextGrad optimizers** | | | | | | | | | | |
| TGD Optimizer | 0.54 | 0.55 | 0.10 | 0.10 | 0.34 | 0.35 | 0.76 | 0.71 | 0.44 | 0.43 |
| ADAS-TG | 0.58 | 0.58 | 0.21 | 0.16 | 0.36 | 0.37 | 0.75 | 0.70 | 0.48 | 0.45 |
| **DSPy optimizers** | | | | | | | | | | |
| Zero-shot MIPROv2 | 0.57 | 0.55 | 0.19 | 0.16 | 0.43 | 0.38 | 0.76 | **0.77** | 0.49 | 0.47 |
| 8-shot MIPROv2 | 0.52 | 0.57 | 0.33 | 0.26 | 0.37 | 0.34 | 0.74 | 0.65 | 0.49 | 0.46 |
| **Meta-optimized optimizers** | | | | | | | | | | |
| `metaTextGrad` | **0.60** | **0.65** | **0.42** | **0.37** | **0.45** | 0.40 | **0.78** | 0.71 | **0.56** | **0.53** |

Table 1: Accuracy (%) of GPT-4o-mini on benchmarks. Bold indicates the best result, and underlined text represents the second-best.

in our experiments, we use GPT-4o-mini for LLM calls within the program, GPT-4o for the MIPROv2 and TGD optimizers, and the o1 model for the structure optimizer and the meta-optimizers.

## 4.2 Main Results

As shown in Table 1, we achieve up to an 11% absolute performance improvement across these datasets, with an average performance significantly surpassing existing optimizers. Our method outperforms both TGD and ADAS-TG, which serve as the base candidates for `metaTextGrad`, across all benchmarks and achieves the best performance on most of them. It's worth noting that the programs optimized by the meta-optimized optimizers also exhibit interesting properties.

(1) The optimized optimizer is **more aligned with specific tasks**. For example, in the BBH Dyck Languages task, the generated program includes components such as n *type analyzer*, and a *stack validator*, which closely match the nature of the task. In contrast, an unaligned optimizer tends to propose more generic and broadly applicable structures.

(2) The optimized optimizer is **more effective in handling finer details**. For instance, in multi-step LLM calls, passing both the overall problem and subproblems to each LLM call helps maintain a global understanding throughout the process. This behavior is more frequently observed in the optimized optimizer, whereas the initial optimizer tends to pass only the subproblems to each subpart.

(3) The meta-optimized optimizer **generally improves efficiency**. Although we allocate six optimization steps per training epoch, we observe that meta-optimized optimizers, due to their stronger task alignment, often achieve significant improvements within the first 1-2 steps. In contrast, other optimizers show more gradual improvements.

Please refer to Appendix E for the optimized programs and Appendix F for the optimized optimizers.

## 4.3 Cost analysis

In this section, we examine the trade-off between effectiveness and computational cost.

First, we analyze the token usage at different levels within a single epoch of meta optimization. This analysis supports our design choice of using models with different capabilities at different levels. As shown in Table 2, the token usage per optimization epoch on the MMLU Abstract Algebra dataset reveals that higher-level components require significantly fewer tokens than lower-level ones. This justifies our hierarchical design. For comparison, a single round of zero-shot CoT requires approximately 140k tokens, indicating that the overall token consumption of our optimization pipeline remains within a reasonable and practical range.

| Level | Tokens |
|---|---|
| Program level | $\sim 400k$ |
| Optimizer level | $\sim 100k$ |
| Meta-optimizer level | $\sim 2.5k$ |

Table 2: Token analysis on Abstract Algebra.

| Model | Performance | Cost |
|---|---|---|
| 0-shot CoT (4o-mini) | 0.05 | 0.14$ |
| Ours (4o-mini) | 0.37 | 0.44$ |
| 0-shot CoT (4o) | 0.18 | 0.52$ |

Table 3: Cost analysis on Dyck Languages.

In addition, we evaluate the cost and performance of the zero-shot CoT approach using both GPT-4o-mini and GPT-4o on the BBH Dyck Languages dataset, and compare them to our optimized approach applied to GPT-4o-mini. As shown in Table 3, our method achieves the best performance on BBH Dyck Languages while incurring a lower cost than GPT-4o. This demonstrates that with effective prompt and structure optimization, a smaller model can outperform the zero-shot performance of a larger model. These findings highlight the practical applicability and scalability of our approach.

### 4.4 Transferability of the optimized optimizer across models and datasets

In this section, we evaluate the transferability of our optimized optimizer across different language models and datasets. As shown in Table 4, our method trained on GPT-4o-mini achieves superior performance compared to unoptimized baselines when evaluated on Claude 3 Haiku. Furthermore, as illustrated in Table 5, our optimizer trained on the GPQA diamond dataset also transfers effectively to the Abstract Algebra dataset. These results demonstrate that our meta-optimized optimizer exhibits strong transferability across models and datasets.

| Method (Claude 3 Haiku) | Dyck Languages | |
|---|---|---|
| | Val | Test |
| Zero-shot CoT | 0.07 | 0.10 |
| **TextGrad optimizers** | | |
| TGD Optimizer | 0.10 | 0.04 |
| ADAS-TG | 0.35 | 0.34 |
| **Optimizers optimized on GPT-4o-mini** | | |
| `metaTextGrad` | 0.32 | 0.35 |

Table 4: Transferability of the optimized optimizer across models.

| Method | Abstract Algebra | |
|---|---|---|
| | Val | Test |
| Zero-shot CoT | 0.74 | 0.70 |
| **TextGrad optimizers** | | |
| TGD Optimizer | 0.76 | 0.71 |
| ADAS-TG | 0.75 | 0.70 |
| **Optimizers optimized on GPQA diamond** | | |
| `metaTextGrad` | 0.78 | 0.77 |

Table 5: Transferability of the optimized optimizer across datasets.

### 4.5 Analysis of the effectiveness of each meta optimizer

In this section, we analyze the contributions of different components of the proposed meta optimizer. As shown in Table 6, we find that all meta optimizers improve performance on the BBH Dyck Languages benchmark. Optimizers optimized using either method outperform the original optimizer. Among them, the meta prompt optimizer achieves the best improvement when applied to ADAS-TG.

| Split | 0-shot CoT | TGD | ADAS-TG | TGD (O) | ADAS-TG (O) | Struct (O) | `metaTextGrad` |
|---|---|---|---|---|---|---|---|
| Val | 0.06 | 0.10 | 0.21 | 0.21 | 0.42 | 0.24 | 0.42 |
| Test | 0.05 | 0.10 | 0.16 | 0.24 | 0.37 | 0.16 | 0.37 |

Table 6: Analysis of the effectiveness of each meta optimizer on Dyck Languages. TGD (O), ADAS-TG (O), and Struct (O) respectively denote the TGD and ADAS-TG optimizers enhanced by the meta prompt optimizer, and the optimizers enhanced by the meta structure optimizer.

Here, `metaTextGrad` produces the same results as the optimized ADAS-TG because the meta optimizer did not find a better option during the meta structure optimization. So, the optimized ADAS-TG was selected as the best result. Notably, the best meta optimizer varies across benchmarks due to task-specific differences. Despite this, all meta optimizers can effectively enhance the performance of a given optimizer.

# 5 Related Work

## 5.1 Prompt optimization

Prompt optimization has proven crucial for improving LLM performance. Initial strategies, such as few-shot learning and in-context learning, demonstrated careful prompt design could significantly boost LLM performance [22]. Techniques like chain-of-thought reasoning [17] and ensemble methods [23] also emerged as popular ways to structure prompts for effective problem-solving. However, hand-crafted approaches are limited in their utility. Efforts to automate prompt optimization have led to the development of gradient-based approaches [24, 25]. These methods, however, require access to model parameters, which limits their application to open-source models.

To address these constraints, alternative approaches have been proposed that leverage LLMs themselves as prompt optimizers. APE [26] was among the first to introduce the concept of automatic instruction generation and selection. [27] introduced the concept of *meta prompt*, demonstrating systematically designing meta-prompts can improve prompt quality. DSPy [4] proposed a systematic approach for optimizing LLM programs, integrating structured optimization techniques.

Another line of research explores optimization methods based on textual feedback. ProTeGi [28] first introduced the concept of textual gradients, highlighting that LLMs themselves can serve as a form of a loss function to guide the improvement of LLM programs. Building on this idea, TextGrad [1] developed a structured textual gradient descent framework, demonstrating its applicability and effectiveness across multiple disciplines and domains. Concurrently, OPTO [29] proposed a related approach in which the optimizer receives an execution trace alongside feedback on the generated output. Semantic gradient descent [30] refines textual gradient descent by enhancing feedback signals.

However, the LLM optimizers proposed in these methods remain fixed during the optimization process, limiting their effectiveness and adaptability. Our approach addresses this limitation by leveraging a meta-optimizer to align LLM optimizers with the task.

## 5.2 Classical meta-learning

Gradient-based meta-learning algorithms such as Model-Agnostic Meta-Learning (MAML) [31], First-Order MAML, Reptile [32], and ANIL [33] cast *learning to learn* as finding a good initialization that can be fine-tuned with only a handful of gradient steps. MAML jointly optimizes across tasks through a bi-level procedure, explicitly encouraging large improvements after one or two inner-loop updates. Reptile shows that a simpler first-order update suffices, while ANIL's ablation studies reveal that the bulk of MAML's gains stem from feature reuse rather than rapid weight adaptation, allowing the inner loop to be removed for all but the task-specific head. Despite their successes, these methods utilize fixed optimizers. In contrast, we focus on optimizing LLM-based optimizers rather than classical ones, and we meta-learn the optimizer so that the optimization strategy is aligned with each new task, rather than merely learning a good initialization.

# 6 Conclusion

In this paper, we propose `metaTextGrad`, a meta-optimization framework that enhances existing LLM optimizers by aligning them more effectively with tasks. Our method introduces two key components: the meta prompt optimizer, which refines optimizer prompts for better task adaptation, and the meta structure optimizer, which determines the optimal combination of different optimizers. By integrating these two components, `metaTextGrad` improves the efficiency of LLM optimizers, leading to better performance across a diverse range of benchmarks.

Looking ahead, there are several promising directions for future research. First, even the meta optimizer we proposed can be optimized. In particular, our meta optimizer is designed by ourselves, instead of learned by data, and there are techniques in the meta learning literature that could be adapted to allow this [34, 35]. Second, future work can explore different ways to parameterize the optimizers. This study primarily focuses on refining optimizer prompts and their composition, but meta-optimizers could also be leveraged to automatically enhance the optimization algorithms employed by existing optimizers. We believe that optimizing LLM optimizers is a crucial step toward further improving the performance and task alignment capabilities of LLM-driven systems and has the potential to provide valuable insights to the research community.

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

# A  Implementation Details

In this section, we provide the implementation of the meta optimizers. The LLMCall component generates responses based on the meta optimizer's prompt. The detailed prompts will be provided in the next section of the appendix.

## A.1  Meta Prompt Optimizer

---
**Algorithm 3** Meta Prompt Optimizer

---
1: **function** $\widehat{M}$.Initialize($\mathcal{D}, M$)
2:   store($\mathcal{D}$)
3:   $M^* \leftarrow M$
4:   $\sigma^* \leftarrow 0$
5:   **return**
6: **end function**
7: **function** $\widehat{M}$.Propose()
8:   question, answer $\sim \mathcal{D}$
9:   proposedOptimizer $\leftarrow$ LLMCall(prompt, $M^*$, question, answer)
10:   **return** proposedOptimizer
11: **end function**
12: **function** $\widehat{M}$.Update($M, \sigma$)
13:   **if** $\sigma > \sigma^*$ **then**
14:     $M^* \leftarrow M$
15:     $\sigma^* \leftarrow \sigma$
16:   **end if**
17:   **return**
18: **end function**
19: **function** $\widehat{M}$.ExtractOptimizedOptimizer()
20:   **return** $M^*$
21: **end function**

---

## A.2  Meta Structure Optimizer

---
**Algorithm 4** Meta Structure Optimizer

---
1: **function** $\widehat{M}$.Initialize($\mathcal{D}, \{M^{(i)}\}_{i=1}^r$)
2:   store($\mathcal{D}, \{M^{(i)}\}_{i=1}^r$)
3:   $M^* \leftarrow$ None
4:   $\sigma^* \leftarrow 0$
5:   **return**
6: **end function**
7: **function** $\widehat{M}$.Propose()
8:   proposedOptimizer $\leftarrow$ LLMCall(prompt, $M^*, \{M^{(i)}\}_{i=1}^r$)
9:   **return** proposedOptimizer
10: **end function**
11: **function** $\widehat{M}$.Update($M, \sigma$)
12:   **if** $\sigma > \sigma^*$ **then**
13:     $M^* \leftarrow M$
14:     $\sigma^* \leftarrow \sigma$
15:   **end if**
16:   **return**
17: **end function**
18: **function** $\widehat{M}$.ExtractOptimizedOptimizer()
19:   **return** $M^*$
20: **end function**

---

## A.3  Meta Optimizer

For the meta optimizer, the execution process consists of two steps: (1) Each optimizer is individually optimized using the meta prompt optimizer. (2) The optimized optimizers are further refined using the meta structure optimizer.

# B  Proof of Theorem 1

**Theorem** (Restatement of Theorem 1). *Let $S_1$ and $S_2$ be two datasets sampled independently from distribution $D$, with sizes $n$ and $m$, respectively. Then, with probability at least $1 - \delta$, the optimizer $\widehat{\theta}_{S_1}$ trained on $S_1$ satisfies:*

$$R_{S_2}(\widehat{\theta}) \leq R(\theta^*) + \sqrt{\frac{2\log(6/\delta)}{n}} + \sqrt{\frac{\log(6/\delta)}{2m}}.$$

*Proof.* According to Hoeffding's inequality, for any $0 < \delta < 1$:

$$\Pr_{S_1 \sim D^n}\left[ |R(\theta) - R_{S_1}(\theta)| > \varepsilon_n(\delta) \right] \leq \delta, \tag{4}$$

where:

$$\varepsilon_n(\delta) = \sqrt{\frac{\log(2/\delta)}{2n}} \tag{5}$$

Using inequality 4, we can bound the population risk of the learned optimizer. With probability at least $1 - \frac{2\delta}{3}$:

$$R(\widehat{\theta}) \leq R_{S_1}(\widehat{\theta}) + \varepsilon_n(\frac{\delta}{3}) \leq R_{S_1}(\theta^*) + \varepsilon_n(\frac{\delta}{3}) \leq R(\theta^*) + 2\,\varepsilon_n(\frac{\delta}{3}), \tag{6}$$

where $\theta^* = \arg\min_\theta R(\theta)$.

With a union bound, with probability at least $1 - \delta$:

$$R_{S_2}(\widehat{\theta}) \leq R(\widehat{\theta}) + \varepsilon_m(\frac{\delta}{3}) \leq R(\theta^*) + 2\,\varepsilon_n(\frac{\delta}{3}) + \varepsilon_m(\frac{\delta}{3}) \tag{7}$$

In conclusion, with probability at least $1 - \delta$ over the draws of $S_1$ and $S_2$:

$$R_{S_2}(\widehat{\theta}) \leq R(\theta^*) + \sqrt{\frac{2\log(6/\delta)}{n}} + \sqrt{\frac{\log(6/\delta)}{2m}} \tag{8}$$

$\square$

# C  Details on Experimental Benchmark Setup

**BBH.**  The BBH task [13, 14] is a challenging benchmark that requires precise reasoning across various tasks. We select two subsets from the BBH benchmark: BBH Word Sorting and BBH Dyck Languages. The BBH Word Sorting task requires the language model to sort a given set of words in order, while the BBH Dyck Languages task involves providing a string composed of various types of brackets and asking the model to determine the characters needed to complete the bracket pairing. We use the same train/validation/test splits as in TextGrad [1] (i.e., 50, 100, and 100 instances for training, validation, and testing) and follow the TextGrad approach by using GPT-4o to evaluate whether the model's output is correct, based on the predicted and ground truth answers.

**MMLU.**  The MMLU benchmark [15] evaluates a model's ability to answer questions across a wide range of scientific disciplines. We selected the MMLU Abstract Algebra dataset and created training, validation, and test sets consisting of 10, 50, and 40 questions, respectively. We assess model accuracy using exact matching, determining correctness by applying a regular expression to check whether the model's response contains "Answer: [A-D]" and matches the ground truth. However, we observed that the DSPy baselines struggle to adhere to this output format. To accommodate this, we use GPT-4o to evaluate the correctness of responses generated by DSPy-based methods.

**GPQA Diamond.**  The GPQA Diamond benchmark [16] evaluates the model's ability to solve graduate-level, Google-proof questions. The benchmark consists of 198 questions, which we split into training, validation, and test datasets containing 30, 100, and 68 questions. Similar to MMLU, we use exact matching to determine response accuracy by applying a regular expression to check whether the model's answer follows the format "Answer: [A-D]" and matches the correct answer. For the DSPy baselines, we use GPT-4o to evaluate the correctness of DSPy-generated responses.

# D   Prompts of Meta Optimizers

In this section, we provide the prompts used in the propose stage of the meta prompt optimizer and the meta structure optimizer. The propose stage is the only part of the meta optimizer that involves an LLM call; all other functions consist of direct numerical updates or result retrieval, as explained in pseudocode in Appendix A.

## D.1   Meta Prompt Optimizer

Below, we present the prompt used for the meta prompt optimizer.

---

**Meta Prompt Optimizer**

\# Task Requirement
A TextGrad optimizer optimizes a TextGrad pipeline so that the pipeline can generate better outputs based on inputs.
A TextGrad pipeline consists of several agents, each of which has a specific role in the optimization process, which is defined by different prompts.
You will be given a general task description of an optimizer and the specific task the pipeline aims to solve.
Your task is to propose an improved optimizer task description so that the optimizer can better optimize the pipeline for the given task.
\# Optimizer Code
Here is the source code of the optimizer: (just for reference)
{optimizer_source_code}
\# The task of the optimizer
Specifically, the pipeline aims to solve this kind of question: {example_set.get_question_type()}.
An example of the task is provided here:
Question: {example_question}
Answer: {example_answer}
The LLM optimizer wants to improve a LLM pipeline to solve such kind of problems.
\# Current Task Description
Here is the current task description of the optimizer, which you can improve: {optimizer_prompt}
\# Your task
You should identify what the optimizer should pay attention to in order to improve the {optimizer_type} of the pipeline for solving the given task.
Conduct a detailed analysis of the given example, and respond in the following format:
```json
"improved_task_description": "...", # Your improved task description for the optimizer
```

---

## D.2   Meta Structure Optimizer

---

**Meta Structure Optimizer**

\# Task Requirement
A TextGrad optimizer optimizes a TextGrad pipeline so that the pipeline can generate better outputs based on inputs.
The structure and prompts of current optimizers are fixed, and you will be given the source code of some optimizers.
Your task is to propose an improved optimizer code to better optimize the pipeline.
This can be achieved by integrating the reference codes of the given different optimizers and merging them into a new optimizer.

---

## Meta Structure Optimizer (Continued)

# Task Details
You will be provided with implementations of several optimizers, each annotated with their respective purposes.

Carefully read through their functions and implementation details. Your task is to integrate the different implementation approaches to provide a more optimized solution.

In this context, the step() function refers to the process of performing a single optimization step.

You may notice that different optimizers are suitable for different purposes and should be applied in a specific order.

For example, if the pipeline aims to optimize the structure, this optimization will overwrite the prompts of each previously optimized component, so the structure optimization should be executed first.

Your implementation of the step() function should take the order into account, and each call to the step() function should only optimize a single aspect of the pipeline because the gradient context is not reusable.

Therefore, you need to implement a logic so that the optimizer can use different optimizing strategies in a sequential manner, with each strategy being applied self.epoch times before moving to the next one.

{optimizer_prompt}

# Code Reminder
You should include the following necessary imports at the beginning of the code:

import inspect
import copy
from typing import Type, List, Union
from collections import defaultdict
from textgrad.variable import Variable
from textgrad import logger
from textgrad.engine import EngineLM
from textgrad.optimizer.optimizer import Optimizer, get_gradient_and_context_text
from textgrad.model import Pipeline
from textgrad.autograd import FormattedLLMCall
from textgrad.config import validate_engine_or_get_default
from textgrad.optimizer.optimizer_prompts import import construct_tgd_prompt, OPTIMIZER_SYSTEM_PROMPT, PIPELINE_SYSTEM_PREFIX, PIPELINE_SYSTEM_SUFFIX

# Note
You are required to build on one of the existing optimizers and improve it by integrating features from the others, instead of starting from scratch.

You should only include the ImprovedOptimizer (inherited exactly from Optimizer) class in the code, and no other classes or functions.

You should implement all the necessary functions and attributes in the ImprovedOptimizer class to ensure that the code can be executed without errors.

You should not modify input signatures of the __init__ and step functions when implementing the ImprovedOptimizer class.

Please output the complete improved Python code, and make sure to call the improved class 'ImprovedOptimizer'; remember to also include all necessary attributes of the class so that the code can be executed without errors.

Do not include any additional comments or unnecessary text in the output.

# E   Optimized Programs

To facilitate the reproducibility of our work, we provide the program generated by the optimizer optimized with `metaTextGrad`.

## E.1   BBH Word Sorting Benchmark

```
BBH Word Sorting

class SimplePipeline(Pipeline):
    task_description = (
        "You will answer a reasoning question. Think step by step. "
        "The last line of your response should be of the following "
        "format: Answer: \$VALUE' "
        "where VALUE is the answer to the question."
    )

    def __init__(self, engine: Union[EngineLM, str] = None, path=None):
        super().__init__(engine=engine, path=path)

        self.planner_prompt = Variable(
            "Create a step by step plan for the question. Provide each
            step on a new line.",
            requires_grad=True,
            role_description="planner prompt"
        )
        self.planner = tg.BlackboxLLM(self.engine, self.planner_prompt)
        self.subsolver_prompt = Variable(
            "Solve this sub-step in detail
            , without giving the final answer.",
            requires_grad=True,
            role_description="sub-step solver prompt"
        )
        self.subsolver = tg.BlackboxLLM(self.engine,
                                        self.subsolver_prompt)

        self.final_prompt = Variable(
            """Combine all reasoning into a coherent final response,
            ending with the format: Answer: $VALUE""",
            requires_grad=True,
            role_description="final answer prompt"
        )
        self.finalsolver = tg.BlackboxLLM(self.engine,
                                          self.final_prompt)
    def _step_split_rule(self, text: str):
        return re.split(r"\n+", text.strip())

    def forward(self, question: Variable) -> Variable:
        plan = self.planner(question)
        steps = Split()(self._step_split_rule, plan)
        sub_solutions = []
        for step in steps:
            sub_solutions.append(self.subsolver(step))
        aggregated_reasoning = Aggregate()(sub_solutions)
        final_answer = self.finalsolver(Aggregate()([question,
        aggregated_reasoning]))
        return final_answer
```

## E.2 BBH Dyck Languages Benchmark

**BBH Dyck Languages**

```python
class SimplePipeline(Pipeline):
    task_description = """You will answer a reasoning question.
    Think step by step. The last line of your response
    should be of the following format:
    'Answer: $VALUE' where VALUE is the answer to the question."""

    def __init__(self, engine: Union[EngineLM, str] = None, path=None):
        super().__init__(engine=engine, path=path)
        if not path:
            self.initial_planner = tg.BlackboxLLM(
                self.engine,
                Variable(
                    """Break down the Dyck sequence validation into
                    detailed steps. Format as numbered steps:
                    1), 2), etc.""",
                    requires_grad=True,
                    role_description="creates detailed analysis plan"
                )
            )

            self.type_analyzer = tg.BlackboxLLM(
                self.engine,
                Variable(
                    """Identify and categorize all bracket types.
                    List each type and its corresponding closing
                    bracket. Format: 'Types:
                    [pairs]'""",
                    requires_grad=True,
                    role_description="analyzes bracket types and pairs"
                )
            )

            self.stack_validator = tg.BlackboxLLM(
                self.engine,
                Variable(
                    """Simulate stack operations for bracket matching.
                    Show stack state after each operation.
                    End with 'Answer:  Valid/Invalid'""",
                    requires_grad=True,
                    role_description="performs stack-based validation"
                )
            )

            self.nesting_analyzer = tg.BlackboxLLM(
                self.engine,
                Variable(
                    """Analyze nesting hierarchy. Check if inner
                    brackets close  before outer brackets.
                    End with 'Answer: Proper/Improper'""",
                    requires_grad=True,
                    role_description="validates nesting hierarchy"
                )
            )
```

```python
        self.depth_checker = tg.BlackboxLLM(
            self.engine,
            Variable(
                """Calculate maximum nesting depth and verify
                balanced structure. End with 'Answer:
                Depth=X,Balanced=Yes/No'""",
                requires_grad=True,
                role_description="checks nesting depth and balance"
            )
        )

        self.sequence_validator = tg.BlackboxLLM(
            self.engine,
            Variable(
                """Validate sequence completeness and correctness.
                End with 'Answer: Complete/Incomplete'""",
                requires_grad=True,
                role_description="validates sequence completeness"
            )
        )

        self.final_evaluator = tg.BlackboxLLM(
            self.engine,
            Variable(
                """Synthesize all analysis results and determine
                if this is a valid Dyck sequence.
                End with 'Answer: Yes/No'""",
                requires_grad=True,
                role_description="makes final validity
                                   determination"
            )
        )

    def forward(self, question: Variable) -> Variable:
        plan = self.initial_planner(question)
        analysis_steps = Split()(lambda x: re.split(r'\d\)', x), plan)

        type_analysis = self.type_analyzer(question)

        stack_validation = self.stack_validator(
            Aggregate()([question, type_analysis])
        )

        nesting_analysis = self.nesting_analyzer(
            Aggregate()([question, stack_validation])
        )

        depth_analysis = self.depth_checker(
            Aggregate()([question, nesting_analysis])
        )

        sequence_validation = self.sequence_validator(
            Aggregate()([question, depth_analysis])
        )
```

```
        final_result = self.final_evaluator(
            Aggregate()([
                question,
                stack_validation,
                nesting_analysis,
                depth_analysis,
                sequence_validation
            ])
        )

        return Extract()(r"Answer: (.+)", final_result)
```

## E.3  GPQA Diamond Benchmark

GPQA Diamond

```
class SimplePipeline(Pipeline):
    task_description = """You will answer a reasoning question.
    Think step by  step. The last line of your response should
    be of the following format: 'Answer: $VALUE' where VALUE is
    the answer to the question."""

    def __init__(self, engine: Union[EngineLM, str] = None, path=None):
        super().__init__(engine=engine, path=path)
        if not path:
            self.executer = textgrad.BlackboxLLM(
                self.engine,
                textgrad.Variable(
                    """You will answer a multiple choice question.
                    Begin by  identifying and listing all key
                    details and constraints from the problem
                    statement. Use explicit reasoning steps to
                    outline the solution, incorporating relevant
                    scientific  principles such as reaction
                    mechanisms or stereochemistry. Verify your
                    initial conclusions by cross-checking each
                    step against the problem's requirements.
                    Consider alternative answers and evaluate
                    why they might be correct or incorrect.
                    Provide a clear justification for your answer
                    choice, and rate your confidence in the final
                    answer on a scale from 1 to 10, explaining
                    your rationale. The last line of your response
                    should be of the following format:
                    'Answer: $VALUE' where VALUE is one of ABCD.""",
                    requires_grad=True,
                    role_description="prompt for the executer,
                    which aims to solve the task"
                )
            )

    def forward(self, question: Variable) -> Variable:
        return self.executer(question)
```

### E.4 MMLU Abstract Algebra Benchmark

**MMLU Abstract Algebra**

```python
class SimplePipeline(Pipeline):
    task_description = """You will answer a reasoning question.
    Think step by step. The last line of your response should
    be of the following format:
    'Answer: $VALUE' where VALUE is the answer to the question."""

    def __init__(self, engine: Union[EngineLM, str] = None, path=None):
        super().__init__(engine=engine, path=path)
        if not path:
            self.executer = textgrad.BlackboxLLM(
                self.engine,
                textgrad.Variable(
                    """You will answer a multiple choice question.
                    Analyze each  statement separately and provide
                    your reasoning. Use clear, logical steps,
                    verifying each sub-component of the problem
                    systematically. Present each statement distinctly
                    using bullet points or numbers, ensuring the final
                    conclusion is  clearly separated from the statement
                    evaluations. Include any assumptions or context
                    necessary for each conclusion. After evaluating
                    each statement, reexamine your conclusions
                    to confirm their correctness. Introduce a summary
                    verification step before concluding. The last line
                    of your response should be of the following format:
                    'Answer: $VALUE' where VALUE is one of ABCD.""",
                    requires_grad=True,
                    role_description="prompt for the executer,
                    which aims to solve the task"
                )
            )

    def forward(self, question: Variable) -> Variable:
        return self.executer(question)
```

## F  Optimized Optimizers

### F.1  Optimized TGD Optimizer Demo

The TGD optimizer is generic, so its optimized program offers only general guidance like "ensuring that every opening bracket has a corresponding closing bracket," without task-specific strategies. In contrast, the optimized TGD aligns closely with the task, emphasizing ideas like "preserving the LIFO (Last In First Out) order of nested structures." Since the optimized TGD optimizer is better aligned with the task, its optimized program also effectively implements the LIFO principle, making it more effective. This is why a task-specific prompt optimizer is necessary.

**TGD Optimizer prompt**

You are part of an optimization system that improves text (i.e., variable). You will be asked to creatively and critically improve prompts, solutions to problems, code, or any other text-based variable.

## F.2 Optimized ADAS-TG Optimizer Demo

The original program contains generic components like a planner, reasoner, and synthesizer, lacking task-specific focus. In contrast, the optimized version adds tailored modules such as a type analyzer, stack validator, and nesting analyzer, aligning closely with BBH Dyck Languages. This illustrates the necessity of a task-specific prompt optimizer.

> **Meta-optimized ADAS-TG Optimizer prompt**
>
> (Some general instructions are omitted here.) Please produce only the code for the pipeline, with the following structural improvements: 1) Implement a stack-based mechanism for tracking open brackets, 2) Create separate components for sequence parsing, bracket matching, and completion generation, 3) Include validation checks for proper nesting and bracket type matching, 4) Ensure systematic handling of different bracket types ([], {}, (), <>), 5) Maintain a hierarchical structure that clearly separates the parsing logic from the completion generation. The pipeline should efficiently handle nested sequences while preserving the LIFO (Last In, First Out) order of brackets.

## G  Performance When Using an Open-source Model as the Optimizer

We further evaluate the generality of our framework by adopting fully open-source models. Specifically, we employ the non-thinking variant of **Qwen3-8B** as the program model and **Qwen3-235B-A22B** as both the optimizer and meta-optimizer. The experiments are conducted on the BBH Dyck Languages benchmark. As shown in Table 7, `metaTextGrad` continues to deliver strong performance.

## H  Performance When Applying to a Challenging Benchmark

To further assess the adaptability of our approach, we evaluate `metaTextGrad` on the **ARC-AGI** benchmark, a task family known for requiring abstract reasoning and compositional generalization. Following common ADAS practice, we sample ARC-AGI instances with grid sizes $\leq 5 \times 5$, comprising 20 training, 30 validation, and 30 test examples. The evaluation reports one-shot success rates, using **Claude 3 Haiku** as the program model and **Claude 3.5 Sonnet** as both optimizer and meta-optimizer. As presented in Table 7, the results demonstrate that `metaTextGrad` remains highly competitive in this more demanding setting.

| Method | Dyck Languages (Qwen models) | | ARC-AGI (Challenging Benchmark) | |
|---|---|---|---|---|
| | **Val** | **Test** | **Val** | **Test** |
| **Vanilla prompting methods** | | | | |
| Zero-shot CoT | 0.27 | 0.27 | 0.27 | 0.23 |
| 8-shot CoT | 0.37 | 0.40 | 0.03 | 0.00 |
| Self-consistency (8) | 0.31 | 0.32 | 0.30 | 0.23 |
| Best of N (8) | 0.39 | 0.41 | 0.27 | 0.20 |
| **TextGrad optimizers** | | | | |
| TGD Optimizer | 0.69 | 0.68 | 0.33 | 0.33 |
| ADAS-TG | 0.32 | 0.34 | 0.28 | 0.26 |
| **DSPy optimizers** | | | | |
| Zero-shot MIPROv2 | 0.59 | 0.50 | 0.30 | 0.23 |
| 8-shot MIPROv2 | 0.57 | 0.51 | 0.33 | 0.03 |
| **Meta-optimized optimizers** | | | | |
| `metaTextGrad` | **0.82** | **0.77** | **0.37** | **0.40** |

Table 7: Validation and test accuracy when using open-source models (Qwen models, Dyck Languages) and evaluating on a challenging benchmark (ARC-AGI). Bold entries denote the best-performing method within each benchmark.

## I  Potential Limitations, Societal Consequences, and Broader Impacts

**Limitations.** While `metaTextGrad` demonstrates stable performance across various models and benchmarks, we acknowledge the following limitations:

(1) Although `metaTextGrad` improves performance on many benchmarks, it may not be effective in scenarios where the base model lacks sufficient task-relevant knowledge or reasoning capabilities. For example, GPT-4o-mini struggles on math competition benchmarks such as AIME 2024, where `metaTextGrad` alone cannot yield significant performance gains.

(2) The meta-optimizer relies on strong instruction-following and problem analysis capabilities. As a result, it currently requires advanced models such as o1 or Claude-3.5-Sonnet. Models like Gemini 1.5 Pro do not yet perform adequately. However, given the rapid progress in model development, we expect that more models will become suitable for the framework in the near future.

**Societal Consequences.** `metaTextGrad` can have meaningful societal consequences.

- **Acceleration of domain-specific AI applications.** By making it easier to adapt LLMs to specific tasks, `metaTextGrad` may accelerate the deployment of more reliable AI solutions in other domains.

- **Risk of automation bias or over-reliance.** As optimizers become more autonomous, users might rely on them without fully understanding their behavior or limitations.

- **Potential misuse for persuasive or manipulative systems.** `metaTextGrad` could be exploited to generate more persuasive outputs.

**Impact Statement.** This paper presents work whose goal is to advance the field of Machine Learning. The paper is solely centered on the methodology itself. How it is applied and for what purpose is entirely up to the users. There are many potential societal consequences of our work, none which we feel must be specifically highlighted here.

