# OpenReview forum: "metaTextGrad: Automatically optimizing language model optimizers"
_NeurIPS.cc/2025/Conference — NeurIPS 2025 poster_

### Official Review · Reviewer_rxek · 2025-06-01

**Clarity:** 3
**Significance:** 3
**Originality:** 4
**Rating:** 5
**Confidence:** 4

**Summary:**

The paper proposes to optimize the prompt optimizers to make optimizer prompts (and code) more contextual and tailored for the task at hand. The authors show that their procedure beats five baselines on three benchmarks. The authors propose Meta Prompt Optimizer to improve optimizer prompts, and Meta Structure Optimizer to edit the multiple optimizers’ code and propose a way to compose them.

**Questions:**

None.

**Ethical Concerns:**

["NO or VERY MINOR ethics concerns only"]

**Limitations:**

The complexity of the setup is quite a lot higher than automatic prompt optimization and way higher than manual prompt optimization, which will limit its application in real life.

**Paper Formatting Concerns:**

None.

**Quality:**

4

**Strengths And Weaknesses:**

Strengths:

[1] The motivation to perform meta-optimization is clearly laid out: the optimizer prompts are too generic, and not problem-specific.

[2] The claims are well supported by the experiments.

[3] The choice of the baselines (CoT, TGD Optimizer, ADAS-TG, Zero-shot MIPROv2, 8-shot MIPROv2) is good. The choice of benchmarks (BBH, MMLU and GPQA) is adequate.

[4] Cost analysis clearly shows the additional token usage by metaTextGrad.

Weaknesses:

[1] The stability of convergence of the meta-optimizer is not discussed.

---

> ### Author Rebuttal · Authors · 2025-07-29
>
> We thank the reviewer for noting that the motivation to perform meta-optimization is clearly laid out, the claims are well supported by the experiments, the choice of baselines is good, the selection of benchmarks is adequate, and the cost analysis clearly shows the additional token usage by metaTextGrad. We will address your concerns in our following response.
>
> > Q1. The stability of convergence of the meta-optimizer is not discussed.
>
> To understand the stability of convergence, we conducted tests on the BBH Dyck Languages benchmark and recorded the training curves for both the inner and outer loops. Since we can only describe the curves in text for the rebuttal, we provide the following summary:
>
> **Outer Loop:**
>
> 0.21, 0.42, 0.42, 0.42, 0.42, 0.42  (Yes, the result converges in the first step.)
>
> **Inner Loop:**
>
> (Outer Loop 0) 0.09, 0.09, 0.09, 0.09, 0.09, 0.14, 0.21, 0.21, 0.21, 0.21, 0.21, 0.21, 0.21, 0.21, 0.21, 0.21
>
> (Outer Loop 1) 0.07, 0.13, 0.13, 0.15, 0.15, 0.15, 0.42, 0.42, 0.42, 0.42, 0.42, 0.42, 0.42, 0.42, 0.42, 0.42
>
> To demonstrate the convergence behavior, we allocated 15 steps to the inner loop and 5 steps to the outer loop in this case. As shown, the training actually converges much earlier. Therefore, in the actual experiments, to conserve resources, we used 6 steps for the inner loop and 2 steps for the outer loop.
>
> > Q2. The complexity of the setup is quite a lot higher than automatic prompt optimization and way higher than manual prompt optimization, which will limit its application in real life.
>
> One of the potential real-world applications of meta-optimization is to train an optimizer that can efficiently adapt to specific tasks, so that it can be used effectively after training. For instance, in agent-based workflows, an agent may need to reliably solve a specific class of problems. In such cases, it is beneficial to first meta-optimize an LLM-based optimizer, so that the agent can save resources when solving a large number of similar tasks in the future.

---

> > ### Comment · Reviewer_rxek · 2025-08-06
> >
> > Dear authors, thank you for addressing my concerns and questions. I have increased my score.

---

> > > ### Author Response · Authors · 2025-08-07
> > > **Thank you**
> > >
> > > We would like to once again express our sincere gratitude for your recognition that the motivation to perform meta-optimization is clearly laid out, the claims are well supported by the experiments, the choice of baselines and benchmarks is appropriate, and the cost analysis clearly illustrates the additional token usage introduced by metaTextGrad.
> > >
> > > We are also truly pleased to hear that our rebuttal has addressed your concerns and questions, and that you will increase your score.
> > >
> > > Best regards!

---

### Official Review · Reviewer_Ss5L · 2025-06-11

**Clarity:** 4
**Significance:** 3
**Originality:** 3
**Rating:** 5
**Confidence:** 4

**Summary:**

LLMs acting as optimizers have shown great potential in diverse tasks, but the optimizers are always fixed and collaborated for general purpose, having not been optimized themselves or customized for specific scenarios. Different optimizers are good at handling different situations and how to automatically choose, or ensemble among them remains a challenge. To address this challenge, this paper proposes metaTextGrad including meta prompt and meta structure optimizer. The former optimizes prompt for each optimizer and the latter performs combination and sequencing to achieve a composite optimizer. Experimental results on relevant benchmarks show the effectiveness of this method.

**Questions:**

* Are all of the baseline methods incorporated in the initial optimizers?
* What about the overhead of the compared baselines, especially two optimizers, TextGrad and DSPy? If the cost is larger than one of the baselines, then the comparison under the exactly same cost is fair. Otherwise, the shown results are somehow unfair.
* The parameters in the algorithms are not given, like the max meta-iterations and max inner-iterations. Maybe curves depicting relation about performance and the number of iterations can provide more straightforward observations, i.e., at which iteration the curve converges. Are the accuracies shown in the table 1 converged?
* See weakness part. If my concerns mentioned could be addressed to some extent, I would consider raising my rating score.

**Ethical Concerns:**

["NO or VERY MINOR ethics concerns only"]

**Final Justification:**

I raise my rating from borderline accept to accept. Authors have addresses my concerns about the details of cost, more clear cases during rebuttal. Optimizing optimizers themselves provide a new ideas aside from prompt engineering. Additionally, the experimental results are solid.

**Limitations:**

yes

**Quality:**

3

**Strengths And Weaknesses:**

### Strengths
* The motivation is very clear and the paper is well-written.
* The observation that the optimizer itself has not been optimized, designed or customized elaborately in terms of specific scenarios is insightful and there is no previous such propose.
* The improvements over initial optimizers are significant on multiple datasets.
### Weaknesses
* BBH includes 20+ subtasks and this paper only selects two datasets. What is the selection criteria? Is this random or selected specifically?
* The paper only analyzes the cost of metaTextGrad but does not compare the cost against the corresponding baselines. What about the overhead of the compared baselines?
* There is no concluded insights from the optimization results. What do the optimized prompts have in common? The specific contents of the intermediate prompts and structures, and the final ones should be analyzed qualitatively, which may provide some rules and insights. How much do the final results deviate from the initial optimizers? And where exactly do they deviate?

---

> ### Author Rebuttal · Authors · 2025-07-29
>
> We thank the reviewer for highlighting the clear motivation, strong writing, insightful observations, and significant improvements across datasets. We will address your concerns in the following response.
>
> >  Q1. BBH includes 20+ subtasks and this paper only selects two datasets. What is the selection criteria? Is this random or selected specifically?
>
> We selected two relatively challenging subtasks, following prior published works such as TextGrad, DARG[1], and this paper[2].
>
> >  Q2. The paper only analyzes the cost of metaTextGrad but does not compare the cost against the corresponding baselines. What about the overhead of the compared baselines? If the cost is larger than one of the baselines, then the comparison under the exactly same cost is fair. Otherwise, the shown results are somehow unfair.
>
> Thanks! The overhead of the compared baselines is indeed smaller, but in optimization, more cost does not necessarily lead to better performance. To illustrate this point, we have added a comparison under exactly the same cost on BBH word sorting. As shown in the table, under the same cost, although the baseline algorithms show some improvement on the validation dataset, their performance on the test dataset shows little improvement or even significant degradation. This suggests that simply increasing training cost may lead to prompt overfitting on the validation dataset.
>
> |Method|Val|Test|
> |---------------------------------|------|------|
> |TGD Optimizer|0.54|0.55|
> |TGD Optimizer (same cost)|0.59|0.56|
> |ADAS-TG|0.58|0.58|
> |ADAS-TG (same cost)|0.58|0.52|
> |Zero-shot MIPROv2|0.57|0.55|
> |Zero-shot MIPROv2 (same cost)|0.62|0.50|
> |8-shot MIPROv2|0.52|0.57|
> |8-shot MIPROv2 (same cost)|0.58|0.56|
> |metaTextGrad|0.60|0.65|
>
> >  Q3. What do the optimized prompts have in common? How much do the final results deviate from the initial optimizers? And where exactly do they deviate?
>
> Thanks for your suggestion! Overall, meta-learning is particularly helpful when the distribution keeps changing. Different tasks correspond to different distributions, and what metaTextGrad essentially does is spend some compute to adapt an existing optimizer to a new distribution (e.g., aligning it with a specific task).  We plan to include insights from the optimization results in the paper and add a figure to illustrate the optimized prompts, helping readers understand what happens after meta-optimization. Since this year's rebuttal is text-only, we provide a brief summary below. The experiment was conducted on BBH Dyck Languages.
>
> **A. Optimized TGD Optimizer Demo**
>
> **TGD Optimizer prompt:**
>
> ```
> You are part of an optimization system that improves text (i.e., variable). You will be asked to creatively and critically improve prompts, solutions to problems, code, or any other text-based variable.
> ```
>
> **Optimized program prompt:**
>
> ```
> You will answer a reasoning question. Think step by step, ensuring logical consistency and accuracy. Explicitly define the role and rules for each type of bracket in the sequence. Verify each step of your reasoning, ensuring that every opening bracket has a corresponding closing bracket.  Compare your predicted sequence with a known ground truth or expected pattern, and explain any discrepancies. If errors are identified, re-evaluate and correct them. Use mental visualization techniques to aid in understanding sequences. Provide a clear and concise explanation. The last line of your response should be of the following format: 'Answer: $VALUE' where VALUE is the answer to the question.
> ```
>
> **Meta-optimized TGD Optimizer prompt:**
>
> ```
> You are part of an optimization system specialized in improving prompts for bracket matching and sequence completion tasks. Your role is to enhance prompts that help solve Dyck language problems, which involve proper nesting and closure of different types of brackets ({}, <>, ()). When improving prompts, focus on these critical aspects: (1) maintaining accurate bracket pair matching, (2) preserving the LIFO (Last In First Out) order of nested structures, (3) handling multiple bracket types simultaneously, and (4) ensuring complete closure of all open brackets. You should critically analyze how the prompts can better guide the model to track open brackets, maintain proper nesting order, and systematically complete sequences. Consider incorporating pattern recognition strategies and explicit validation rules in the improved prompts. Your improvements should lead to more reliable and accurate bracket sequence completions.
> ```
>
> **Optimized program prompt:**
>
> ```
> You will answer a reasoning question. Focus on identifying and providing only the missing closing symbols needed to complete the sequence. Use a stack-based approach to track opening and closing symbols, ensuring proper nesting and closure. Explicitly push each opening symbol onto the stack and pop it when a corresponding closing symbol is encountered. After processing each symbol, describe the current state of the stack, focusing on unmatched opening symbols. Ensure that the order of closing symbols matches the reverse order of unmatched opening symbols as they appear in the stack. Verify the sequence by checking each symbol's nesting and closure order. Highlight critical decision points, such as when to pop from the stack or add closing symbols. Use visual aids if necessary to represent the stack state and sequence operations. After completing your analysis, perform a verification check by comparing the predicted closing symbols with the expected output. If discrepancies are detected, revisit previous steps to self-correct. Conclude with a summary of the final state of the stack and confirm that all brackets are matched. The last line of your response should be of the following format: 'Answer: $VALUE' where VALUE is the answer to the question.
> ```
>
> **Explanation:**
>
> Since the TGD optimizer is relatively generic, the optimized program, while mentioning something like ```"ensuring that every opening bracket has a corresponding closing bracket,"``` only discusses the approach in general terms and does not provide any specific or targeted guidance.
>
> Compared to the TGD optimizer, the optimized optimizer is better aligned with the task requirements, highlighting key problem-solving ideas such as ```"preserving the LIFO (Last In First Out) order of nested structures."``` Since the optimized TGD optimizer is better aligned with the task, its optimized program also effectively implements the LIFO principle, making it more effective.
>
> **B. Optimized ADAS-TG Optimizer Demo**
>
> **ADAS-TG Optimizer prompt:**
>
> ```
> (Some general instructions are omitted here.) Please produce only the code for the pipeline, with a more systematic or hierarchical structure if possible.
> ```
>
> **Meta-optimized ADAS-TG Optimizer prompt:**
>
> ```
> (Some general instructions are omitted here.)  Please produce only the code for the pipeline, with the following structural improvements: 1) Implement a stack-based mechanism for tracking open brackets, 2) Create separate components for sequence parsing, bracket matching, and completion generation, 3) Include validation checks for proper nesting and bracket type matching, 4) Ensure systematic handling of different bracket types ([], {}, (), <>), 5) Maintain a hierarchical structure that clearly separates the parsing logic from the completion generation. The pipeline should efficiently handle nested sequences while preserving the LIFO (Last In, First Out) order of brackets.
> ```
>
> **Explanation:**
>
> We will include the program produced by the original ADAS-TG, while the program optimized by the meta-optimized ADAS-TG can be found in Appendix pages 18–20. In brief, the original ADAS-TG-generated program consists of components such as a planner, reasoner, and synthesizer, all of which are quite generic and do not incorporate task-specific characteristics. In contrast, the optimized ADAS-TG-generated program includes components such as a planner, type analyzer, stack validator, nesting analyzer, depth checker, sequence validator, and evaluator, which are tailored specifically to the characteristics of BBH Dyck Languages.
>
> >  Q4. Are all of the baseline methods incorporated in the initial optimizers?
>
> Since the codebases of different engines are completely incompatible, directly optimizing different optimizers together is challenging in practice. As noted in line 219 of our paper, in our experiments, we used the TextGrad and ADAS-TG as the base candidates.
>
> >  Q5. The parameters in the algorithms are not given, like the max meta-iterations and max inner-iterations. Maybe curves depicting relation about performance and the number of iterations can provide more straightforward observations, i.e., at which iteration the curve converges. Are the accuracies shown in the table 1 converged?
>
> To understand the relationship between performance and the number of iterations, we conducted tests on BBH Dyck Languages and recorded the training curves for both the inner and outer loops. Since we can only describe the curves in text, we provide the following summary:
>
> **Outer Loop:**
>
> 0.21, 0.42, 0.42, 0.42, 0.42, 0.42  (Yes, the result converges in the first step.)
>
> **Inner Loop:**
>
> (Outer Loop 0) 0.09, 0.09, 0.09, 0.09, 0.09, 0.14, 0.21, 0.21, 0.21, 0.21, 0.21, 0.21, 0.21, 0.21, 0.21, 0.21
>
> (Outer Loop 1) 0.07, 0.13, 0.13, 0.15, 0.15, 0.15, 0.42, 0.42, 0.42, 0.42, 0.42, 0.42, 0.42, 0.42, 0.42, 0.42
>
> To demonstrate the convergence behavior, we allocated 15 steps to the inner loop and 5 steps to the outer loop in this case. As shown, the training actually converges much earlier. Therefore, in the actual experiments, to conserve resources, we used 6 steps for the inner loop and 2 steps for the outer loop.
>
> ---
> [1] Zhang, Z., et al. DARG: Dynamic Evaluation of Large Language Models via Adaptive Reasoning Graph. In NeurIPS 2024.
>
> [2] Akyürek, E., et al.The Surprising Effectiveness of Test-Time Training for Few-Shot Learning. In ICML 2025.

---

> > ### Comment · Reviewer_Ss5L · 2025-08-04
> > **Response to Authors' rebuttal**
> >
> > Thanks for the detailed rebuttal! The cost analysis shows overfitting results of baseline methods, under same cost. However, it is important to analyze at the convergence step since different methods may converge at different steps, also bringing different overheads, I was not convinced by the given results since they are a bit one-sided. Some methods may not need so many steps and then it will not overfit the validation dataset. The case study is in depth and it can be incorporated in the next version.

---

> > > ### Author Response · Authors · 2025-08-05
> > > **Follow-up on Cost Analysis**
> > >
> > > Dear reviewer Ss5L,
> > >
> > > We kindly ask if the additional cost analysis and baseline performance evaluation have addressed your concerns. Should there remain any unresolved issues or aspects of the paper that you feel need further clarification, please don’t hesitate to let us know. We are eager and open to further discussions to ensure clarity and full understanding.

---

> > > > ### Comment · Reviewer_Ss5L · 2025-08-05
> > > > **Response to Authors' rebuttal**
> > > >
> > > > The table showing results at different steps does not provide those of your method, metaTextGrad. Are there any explanations?

---

> ### Author Response · Authors · 2025-08-04
>
> Thank you very much for your response! We are glad to see that you found the case study to be in-depth, and we will include the case study in the revised paper.
>
> We agree with your point that it is important to analyze at the convergence step, as different methods may converge at different points. Therefore, we have increased the density of data points to show the full optimization process of each method until convergence. For the TextGrad baselines, we adjust the optimization steps. For the DSPy baselines, we adjust the number of candidates and number of trials.
>
> As shown in the table, although the baseline methods converge at different steps, their performance consistently remains significantly lower than that of metaTextGrad.
>
> | Method |Val0 | Test0| Val1  | Test1 | Val2 | Test2 | Val3 | Test3 | Val4 | Test4 | Val5 | Test5 | Best Val | Best Test|
> |-|-|-|-|-|-|-|-|-|-|-|-|-|-|-|
> | TGD Optimizer    |0.46|0.55| 0.51 | 0.53  |  0.54  |  0.55     |  0.54      |   0.55     |   0.59   |  0.56     |   0.59   |  0.56     | 0.59| 0.56|
> | ADAS-TG          |0.46|0.55| 0.53 | 0.57 | 0.56    |  0.61    |    0.58   |    0.58    |   0.58    |   0.58     |     0.58  |    0.52    | 0.58| 0.61|
> | Zero-shot MIPROv2 |0.46|0.55| 0.52 | 0.55  |  0.55    |  0.54    |  0.57    |    0.57    |   0.62   |   0.53    |    0.62   |  0.50     | 0.62| 0.57 |
> | 8-shot MIPROv2    |0.46|0.55| 0.49 | 0.53  |  0.51   |   0.59   |  0.52     |    0.57    |   0.56   |    0.53   |   0.58    |  0.56     | 0.58| 0.59|
>
> Since we can only provide text here, we have presented the data in a table format. However, in the revised paper, we will present this experiment as a figure to help readers better understand the convergence behavior of the different baselines.

---

> ### Author Response · Authors · 2025-08-05
>
> Thank you for your prompt response! For each of the meta prompt optimizer and meta structure optimizer, performance typically converges within the first 1–2 steps, so there's no need to present tabular data for them.
>
> However, we will provide a table demonstrating how the meta prompt optimizer and meta structure optimizer sequentially optimize to ultimately achieve good performance.
>
> | Split                     | Before Optimization | TGD  | TGD (O) | ADAS-TG | ADAS-TG (O) | metaTextGrad |
> |--------------------------|------------|------|---------|---------|-------------|--------------|
> | BBH Dyck Languages Val   | 0.06       | 0.10 | 0.21    | 0.21    | 0.42        | 0.42         |
> | BBH Dyck Languages Test  | 0.05       | 0.10 | 0.24    | 0.16    | 0.37        |0.37         |
> | BBH Word Sorting Val     | 0.46       | 0.54 | 0.60    | 0.58   | 0.60        | 0.60         |
> | BBH Word Sorting Test    | 0.55       | 0.55 | 0.59    | 0.58    | 0.55        | 0.65         |
> | **Average**              | **0.28**   | **0.32** | **0.41** | **0.38** | **0.48**    |  **0.51**     |
>
> In the table, the meta prompt optimizer first attempts to optimize the TGD and ADAS-TG optimizers (shown in the 3rd and 5th columns, respectively), followed by the application of the meta structure optimizer, which results in metaTextGrad (shown in the 6th column).

---

> > ### Comment · Reviewer_Ss5L · 2025-08-06
> > **Response to Authors' Rebuttal**
> >
> > The number of steps is one aspect to analyze the cost. And the cost analysis (tokens and cost) in the original paper does not show the comparison with some baselines. Could you show more in-depth results, mainly compared with TGD Optimizer, ADAS-TG, MIPROv2, etc.?

---

> > > ### Author Response · Authors · 2025-08-07
> > >
> > > Thank you very much for your feedback! We appreciate your perspective. Comparing the cost is indeed meaningful. Therefore, we provide the cost incurred by metaTextGrad and each baseline at different steps. Since it's not possible to include plots here, we will use tables to present the rounded cost estimates. In the revision, we will include a cost analysis section with a Pareto frontier plot of cost vs. performance to help readers better understand the trade-offs from a cost perspective.
> > >
> > > | Cost |Val ($0.1) | Test ($0.1) | Val ($0.2)  | Test  ($0.2) | Val  ($0.3) | Test  ($0.3) | Val  ($0.5) | Test  ($0.5) | Val  ($5) | Test  ($5) | Val  ($35) | Test  ($35) | Best Val | Best Test|
> > > |-|-|-|-|-|-|-|-|-|-|-|-|-|-|-|
> > > | TGD |0.46|0.55| 0.51 | 0.53  |  0.54  |  0.55     |  0.54      |   0.55     |   0.59   |  0.56     |   0.59   |  0.56     | 0.59| 0.56|
> > >
> > >
> > > | Cost |Val ($0.1) | Test ($0.1) | Val ($1)  | Test ($1) | Val ($2) | Test ($2) | Val ($5) | Test ($5) | Val ($15) | Test ($15) | Val ($35) | Test ($35) | Best Val | Best Test|
> > > |-|-|-|-|-|-|-|-|-|-|-|-|-|-|-|
> > > | ADAS-TG          |0.46|0.55| 0.53 | 0.57 | 0.56    |  0.61    |    0.58   |    0.58    |   0.58    |   0.58     |     0.58  |    0.52    | 0.58| 0.61|
> > >
> > > | Cost |Val ($0.1) | Test ($0.1) | Val ($1)  | Test ($1) | Val ($2) | Test ($2) | Val  ($3) | Test  ($3) | Val  ($15) | Test  ($15) | Val  ($35) | Test  ($35) | Best Val | Best Test|
> > > |-|-|-|-|-|-|-|-|-|-|-|-|-|-|-|
> > > | Zero-shot MIPROv2 |0.46|0.55| 0.52 | 0.55  |  0.55    |  0.54    |  0.57    |    0.57    |   0.62   |   0.53    |    0.62   |  0.50     | 0.62| 0.57 |
> > >
> > > | Cost |Val ($0.1)  | Test ($0.1) | Val ($1) | Test ($1) | Val ($2) | Test ($2) | Val ($4) | Test ($4) | Val ($15) | Test ($15) | Val ($35) | Test ($35) | Best Val | Best Test|
> > > |-|-|-|-|-|-|-|-|-|-|-|-|-|-|-|
> > > | 8-shot MIPROv2    |0.46|0.55| 0.49 | 0.53  |  0.51   |   0.59   |  0.52     |    0.57    |   0.56   |    0.53   |   0.58    |  0.56     | 0.58| 0.59|
> > >
> > > In comparison, metaTextGrad reaches a validation score of 0.60 and a test score of 0.65 at a cost of $38, a performance level unattainable by the baselines at any cost.

---

> > > > ### Comment · Reviewer_Ss5L · 2025-08-08
> > > > **Response to Authors' rebuttal**
> > > >
> > > > I have a question about the table showing cost of each step on val and test sets. Intuitively, the prompt of 8-shot MIPROv2 is much longer than zero-shot MIPROv2, why do they cost almost the same? Even, 8-shot one cost less on some steps.

---

> ### Author Response · Authors · 2025-08-08
>
> Thank you for your thoughtful question! Recall that our aim is to perform a cost analysis, so the purpose of setting up different steps is to align the costs as much as possible, rather than the steps themselves. Please note that Zero-shot MIPROv2, 8-shot MIPROv2, TGD, and ADAS-TG do not share the same parameter configurations.
>
> That said, in the fourth val/test group, which comes directly from the original paper, Zero-shot MIPROv2 and 8-shot MIPROv2 do use the same number of steps. So the natural question is: why are their costs so close (specifically, `$3` and `$4`), despite the “8-shot” setup seemingly implying much more computation? This is due to the specific implementation details of MIPROv2.
>
> 1. Zero-shot MIPROv2 still needs to generate the same number of demos.
> In MIPROv2, the instruction proposal phase depends on demos, so demo generation is unavoidable even in the zero-shot case.
>
> 2. 8-shot MIPROv2 does not imply that 8 demos are used every time.
> Rather, it means that 8 candidate demos are generated, and in each iteration, **only one demo** is sampled using ```trial.suggest_categorical```.
> A more accurate name might be ```MIPROv2 with 8 candidate demos```.
>
> We sincerely apologize if the term "8-shot MIPROv2" caused any misunderstanding. We will revise this name in the revised paper. Thank you again for raising this important point!

---

> > ### Comment · Reviewer_Ss5L · 2025-08-08
> > **Response to Authors' rebuttal**
> >
> > Thanks for explanation. I do not have other concerns and I suggest all of these experimental results be incorporated in the next version, including cost analysis and in-depth case study, etc. I will raise my overall rating to 5.

---

> > > ### Author Response · Authors · 2025-08-08
> > > **Thank you**
> > >
> > > We sincerely appreciate your recognition that the motivation is very clear, the paper is well-written, our core observation is both insightful and novel, and that the improvements over the initial optimizers are significant across multiple datasets.
> > >
> > > We are very glad to see that all of your concerns have been addressed. In the next version of the paper, we will include all of these experimental results, including the cost analysis and in-depth case study.
> > >
> > > Best regards!

---

### Official Review · Reviewer_V1eb · 2025-07-02

**Clarity:** 1
**Significance:** 3
**Originality:** 3
**Rating:** 4
**Confidence:** 5

**Summary:**

This paper introduces metaTextGrad, a novel meta-optimizer designed to enhance existing prompt optimizers for large language models (LLMs). The core premise is that current prompt optimizers are fixed and lack adaptability to specific tasks. metaTextGrad aims to address this by leveraging meta-learning to align the optimizer with the given task, thereby improving prompt optimization performance.

**Questions:**

1. How do the authors select the benchmarks? For example, it is mentioned in the paper that BBH is a challenging dataset. But in some recent based model (e.g., Qwen 3), they can easily achieve a performance of 86+.

2. Why do the authors use a different model for baselines and the proposed method?

3. What is the performance when using an open-source model, like Qwen or Llama, as the optimizer?

**Ethical Concerns:**

["NO or VERY MINOR ethics concerns only"]

**Final Justification:**

The author's response addresses my concerns well, except for the second weakness. The only weakness does not matter the contribution of the paper, so I raise my score to 4.

**Limitations:**

yes

**Quality:**

2

**Strengths And Weaknesses:**

**Strengths**:

1. The paper focuses on prompt optimization, a crucial and practical aspect of developing robust LLM-based applications. The exploration of LLMs as optimizers is an important, underexplored area.

2. Applying meta-learning principles to the domain of prompt optimization is an interesting and potentially impactful approach.

**Weaknesses**:

1. The motivation is not clear. The authors claim that the existing prompt optimizer is fixed and lacks a process to align with the specific task. However, they lack the explanation about what benefit can be obtained when adapted to a specific task. This makes the claim and motivation of the paper weak.

2. Following from the first point, it is not clear why a task-specific prompt optimizer is necessary. Actually, the optimization on the specific task highly depends on the task goal, which can be reflected by the design of the evaluation, like the evaluation metric or LLM-evaluator. In DSPy or TextGrad, they take the evaluation metrics as input, dynamically adapting to the specific task goal. This undermines the foundational claim and motivation of the work.

3. The paper structure is disjointed. For example, the motivation is presented in several sections separately, making it difficult to understand the main contribution.

4. The statement "Theorem 1 implies that aligning the optimizer with the task through meta-learning can be highly effective" in Section 3.1 lacks the necessary rigor. "Highly effective" is a subjective claim, and it is not immediately apparent from the presented theoretical insight how significant or quantifiable this effectiveness would be. The theoretical contribution should provide more concrete implications.

5. Figures 1 and 2 are difficult to read. Improving their clarity and readability is essential for conveying key information.

6. It is not fair to use GPT-4o for baselines, but GPT-o1 for the proposed method.

---

> ### Author Rebuttal · Authors · 2025-07-29
>
> We thank the reviewer for recognizing the potential of LLM optimizers and the promise of applying meta-learning. We will address your concerns in the following response.
>
> > Q1. The authors claim that the existing prompt optimizer is fixed and lacks a process to align with the specific task. However, they lack the explanation about what benefit can be obtained when adapted to a specific task. It is not clear why a task-specific prompt optimizer is necessary.
>
> Thanks! We plan to include insights from the optimization results and add a figure to illustrate the optimized prompts, helping readers understand what happens after meta-optimization, what benefits can be obtained when adapted to a specific task, and why a task-specific prompt optimizer is necessary. Overall, meta-learning is particularly helpful when the distribution keeps changing. Different tasks correspond to different distributions, and what metaTextGrad essentially does is spend some compute to adapt an existing optimizer to a new distribution (e.g., aligning it with a specific task).
>
> We support this with experiments on BBH Dyck Languages. Since this year's rebuttal is text-only, we provide a brief summary below.
>
> **A. Optimized TGD Optimizer Demo**
>
> **TGD Optimizer prompt:**
>
> ```
> You are part of an optimization system that improves text (i.e., variable). You will be asked to creatively and critically improve prompts, solutions to problems, code, or any other text-based variable.
> ```
>
> **Optimized program prompt:**
>
> ```
> You will answer a reasoning question. Think step by step, ensuring logical consistency and accuracy. Explicitly define the role and rules for each type of bracket in the sequence. Verify each step of your reasoning, ensuring that every opening bracket has a corresponding closing bracket. Compare your predicted sequence with a known ground truth or expected pattern, and explain any discrepancies. If errors are identified, re-evaluate and correct them. Use mental visualization techniques to aid in understanding sequences. Provide a clear and concise explanation. The last line of your response should be of the following format: 'Answer: $VALUE' where VALUE is the answer to the question.
> ```
>
> **Meta-optimized TGD Optimizer prompt:**
>
> ```
> You are part of an optimization system specialized in improving prompts for bracket matching and sequence completion tasks. Your role is to enhance prompts that help solve Dyck language problems, which involve proper nesting and closure of different types of brackets ({}, <>, ()). When improving prompts, focus on these critical aspects: (1) maintaining accurate bracket pair matching, (2) preserving the LIFO (Last In First Out) order of nested structures, (3) handling multiple bracket types simultaneously, and (4) ensuring complete closure of all open brackets. You should critically analyze how the prompts can better guide the model to track open brackets, maintain proper nesting order, and systematically complete sequences. Consider incorporating pattern recognition strategies and explicit validation rules in the improved prompts. Your improvements should lead to more reliable and accurate bracket sequence completions.
> ```
> **Optimized program prompt:**
>
> ```
> You will answer a reasoning question. Focus on identifying and providing only the missing closing symbols needed to complete the sequence. Use a stack-based approach to track opening and closing symbols, ensuring proper nesting and closure. Explicitly push each opening symbol onto the stack and pop it when a corresponding closing symbol is encountered. After processing each symbol, describe the current state of the stack, focusing on unmatched opening symbols. Ensure that the order of closing symbols matches the reverse order of unmatched opening symbols as they appear in the stack. Verify the sequence by checking each symbol's nesting and closure order. Highlight critical decision points, such as when to pop from the stack or add closing symbols. Use visual aids if necessary to represent the stack state and sequence operations. After completing your analysis, perform a verification check by comparing the predicted closing symbols with the expected output. If discrepancies are detected, revisit previous steps to self-correct. Conclude with a summary of the final state of the stack and confirm that all brackets are matched. The last line of your response should be of the following format: 'Answer: $VALUE' where VALUE is the answer to the question.
> ```
>
> **Explanation:**
>
> The TGD optimizer is generic, so its optimized program offers only general guidance like ```"ensuring that every opening bracket has a corresponding closing bracket,"``` without task-specific strategies. In contrast, the optimized TGD aligns closely with the task, emphasizing ideas like ```"preserving the LIFO (Last In First Out) order of nested structures."``` Since the optimized TGD optimizer is better aligned with the task, its optimized program also effectively implements the LIFO principle, making it more effective. This is why a task-specific prompt optimizer is necessary.
>
> **B. Optimized ADAS-TG Optimizer Demo**
>
> **ADAS-TG Optimizer prompt:**
>
> ```
> (Some general instructions are omitted here.) Please produce only the code for the pipeline, with a more systematic or hierarchical structure if possible.
> ```
>
> **Meta-optimized ADAS-TG Optimizer prompt:**
>
> ```
> (Some general instructions are omitted here.) Please produce only the code for the pipeline, with the following structural improvements: 1) Implement a stack-based mechanism for tracking open brackets, 2) Create separate components for sequence parsing, bracket matching, and completion generation, 3) Include validation checks for proper nesting and bracket type matching, 4) Ensure systematic handling of different bracket types ([], {}, (), <>), 5) Maintain a hierarchical structure that clearly separates the parsing logic from the completion generation. The pipeline should efficiently handle nested sequences while preserving the LIFO (Last In, First Out) order of brackets.
> ```
>
> **Explanation:**
>
> We will include the program produced by the original ADAS-TG, while the program optimized by the meta-optimized ADAS-TG can be found in Appendix pages 18–20. The original program contains generic components like a planner, reasoner, and synthesizer, lacking task-specific focus. In contrast, the optimized version adds tailored modules such as a type analyzer, stack validator, and nesting analyzer, aligning closely with BBH Dyck Languages. This illustrates the necessity of a task-specific prompt optimizer.
>
> > Q2. The motivation is presented in several sections separately. Figures 1 and 2 are difficult to read. Improving their clarity and readability is essential for conveying key information.
>
> Thank you very much! We agree with your suggestion. We plan to reorganize Sections 3.2 to 3.4 and consolidate the motivation into a single subsection for greater clarity. Additionally, one issue with Figure 2 is that it doesn’t clearly convey the overall pipeline of metaTextGrad. To address this, we will incorporate concrete examples related to the motivation (as you suggested in Q1) into Figure 2. We believe that providing specific examples will help readers better understand both the necessity and the process of meta-optimization, thereby improving the clarity of the section and the figures.
>
> > Q3. "Highly effective" is a subjective claim. The theoretical contribution should provide more concrete implications.
>
> Thanks! We will revise the phrasing of "Highly effective". Our theoretical contribution lies in highlighting the **necessity** of performing meta-optimization. Specifically, an effectively meta-optimized optimizer can have theoretical performance guarantees, whereas an LLM optimizer without meta-optimization lacks such guarantees and may perform arbitrarily poorly.
>
> > Q4. How do the authors select the benchmarks?
>
> Our benchmarks (BBH, MMLU, and GPQA) follow the selection made in prior published works such as TextGrad, DARG[1], REVOLVE[2], and this paper[3].
>
> > Q5. What is the performance when using an open-source model as the optimizer?
>
> We conducted experiments entirely using open-source models on BBH Dyck Languages. Specifically, we used the non-thinking version of Qwen3-8B as the program model, and the non-thinking version of Qwen3-235B-A22B as the optimizer and meta-optimizer. metaTextGrad still significantly outperforms other baseline methods.
>
> |**Method**|**Dyck Languages**||
> |-|-|-|
> ||**Val**|**Test**|
> |**Vanilla prompting methods**||
> |Zero-shot CoT|0.27|0.27|
> |8-shot CoT|0.37|0.40|
> |Self-consistency (8)|0.31|0.32|
> |Best of N (8)|0.39|0.41|
> |**TextGrad optimizers**||
> |TGD Optimizer|0.69|0.68|
> |ADAS-TG|0.32|0.34|
> |**DSPy optimizers**||
> |Zero-shot MIPROv2|0.59|0.50|
> |8-shot MIPROv2|0.57|0.51|
> |**Meta-optimized optimizers**||
> |metaTextGrad|**0.82**|**0.77**|
>
> > Q6. Why do the authors use a different model for baselines and the proposed method?
>
> We use the o1 model only as the meta-optimizer, while the optimizer itself uses the same model as in the baselines. As shown in Section 4.3, the overhead introduced by the o1 model is minimal compared to the overall cost of the optimization process. As noted in lines 208–213 of the paper, we aim to adopt a hierarchical structure, where a small fixed cost from a model like o1 can significantly improve the quality of the LLM optimizer and better align it with the task.
>
> However, we would like to emphasize that **our method remains effective when the optimizer and meta-optimizer use the same model**. In the experiments in ```Q5```, the same model is used for both the optimizer and meta-optimizer, yet metaTextGrad still significantly outperforms the baselines.
>
> ---
> [1]Zhang, Z., et al. DARG. In NeurIPS 2024.
>
> [2]Zhang, P., ei al. Revolve. In ICML 2025.
>
> [3]Akyürek, E., et al. The Surprising Effectiveness of Test-Time Training for Few-Shot Learning. In ICML 2025.

---

> > ### Comment · Reviewer_V1eb · 2025-08-07
> > **Response to Author's Rebuttal**
> >
> > Thanks for the detailed rebuttal. It addressed some of my concerns, but the key points in weaknesses 1 and 2 haven't been resolved.
> >
> > 1. In the authors' response, meta-learning is beneficial when the distribution is changing. This is quite a generic statement in traditional meta-learning (before 2023). In this scenario, what does "distribution" refer to?
> > 2. The question in weakness 2 has not been answered. Why is a task-specific optimizer necessary instead of a simpler task-specific evaluator?
> >
> > My final score on this paper will depend on the authors' response.

---

> > > ### Author Response · Authors · 2025-08-08
> > > **Official Reply for Reviewer  V1eb (1/2)**
> > >
> > > Thank you very much for taking the time to review our rebuttal. We are pleased to see that it has addressed several of your concerns. We will begin by elaborating on the motivation behind metaTextGrad, and then proceed to address your remaining concerns in the following response.
> > >
> > > > Motivation
> > >
> > > First, existing optimizers such as TextGrad and DSPy are manually designed by humans with the goal of performing well across a broad distribution of tasks, and they indeed demonstrate strong average performance.
> > >
> > > For example, the prompt used by the TextGrad TGD optimizer is as follows:
> > >
> > > ```
> > > You are part of an optimization system that improves text (i.e., variable). You will be asked to creatively and critically improve prompts, solutions to problems, code, or any other text-based variable.
> > > ```
> > >
> > > As can be seen, this prompt is indeed highly general, with phrasing such as `improve prompts, solutions to problems, code, or any other text-based variable`.
> > >
> > > **However, in some cases, such general-purpose prompt fails to effectively optimize model performance. While the optimizers in both TextGrad and DSPy can obtain learning signals from task-specific evaluators, these signals tend to be noisy and sparse.**
> > >
> > > First, when feedback is provided solely in the form of scalar scores rather than textual guidance, it becomes sparse, making optimization substantially more challenging. Second, although optimizers such as TextGrad's TGD can accept textual gradient feedback, such feedback is often highly noisy, making it difficult for the optimizer to learn effectively. For example, the evaluator feedback received by the TGD optimizer may look like the following:
> > >
> > > ```
> > > To improve the prompt for the executer and enhance the objective function, consider the following feedback:\n\n1. **Explicit Criteria Definition**: The prompt should explicitly instruct the executer to define the criteria for optical activity at the beginning of the response. This can prevent ambiguity and ensure that the executer uses the correct scientific principles. For example, the prompt could include a directive to \"List the criteria for optical activity before analyzing each compound.\"\n\n2. **Data Verification Directive**: Incorporate a step in the prompt that requires the executer to verify the input data against reliable sources. This could be phrased as \"Cross-check the properties of each compound with a trusted chemical database before proceeding with the analysis.\"\n\n3. **Structured Logical Reasoning**: Encourage a structured approach to reasoning by breaking down the analysis into distinct steps. The prompt could suggest a format like \"For each compound, first identify chiral centers, then assess symmetry, and finally determine optical activity.\"\n\n4. **Cross-Referencing Encouragement**: Add a directive for the executer to compare its conclusions with known examples or established knowledge. This could be included as \"After determining optical activity, compare your findings with similar known compounds to validate your conclusion.\"\n\n5. **Error Analysis Mechanism**: Introduce a step where the executer reflects on potential errors in its reasoning. The prompt could include \"After reaching a conclusion, review your analysis for any assumptions or skipped steps that could lead to errors.\"\n\n6. **Feedback Loop Integration**: Suggest a mechanism for the executer to learn from past mistakes. This could be phrased as \"If you encounter discrepancies in your conclusion, note them for future reference to improve accuracy.\"\n\nBy incorporating these elements into the prompt, the executer can be guided to produce more accurate and reliable responses, thereby improving alignment with the ground truth answer and enhancing the objective function.
> > > ```
> > >
> > > Here, the evaluator feedback includes suggestions such as `List the criteria for optical activity before analyzing each compound`, which are specific to a single problem instance. Such feedback is clearly noisy. Since a generic optimizer relies solely on feedback to guide its updates, it is likely to incorporate such suggestions into the optimized LLM program prompt, which can be detrimental to the overall performance of the program.

---

> > > ### Author Response · Authors · 2025-08-08
> > >
> > > We hope our response has addressed your concerns. If there are any remaining concerns, please do not hesitate to let us know. We are eager to engage in further discussion to ensure clarity.

---

> ### Author Response · Authors · 2025-08-04
>
> Dear reviewer  V1eb,
>
> We would be deeply grateful if you could kindly find the time to review our rebuttal and the other reviewers' valuable comments. Should there remain any unresolved concerns of the paper you feel need further explanation, please let us know. We are eager and open to further discussions to ensure clarity and comprehension.
>
> If you find that our responses have adequately addressed the initial concerns, we kindly request you to consider revisiting the rating of the paper. Your insights and evaluation are crucial to us, and we sincerely hope our efforts align with your expectations.

---

> ### Author Response · Authors · 2025-08-08
> **Official Reply for Reviewer V1eb (2/2)**
>
> Yet, in reality, we can choose to let the LLM optimizer adapt to a specific task distribution in order to achieve better performance. This is because **if the distribution of programs generated by the task-specific optimizer is more aligned with the task requirements, the difficulty of finding the optimal program will be significantly reduced, and the impact of noise in the evaluator’s signal will be greatly mitigated, even when the feedback is noisy.**
>
> We illustrate this using the BBH Dyck Languages task as an example, showing that aligning the LLM optimizer to a specific task distribution can lead to improved performance. For instance, consider the following task-specific optimizer prompt:
>
> ```
> You are part of an optimization system specialized in improving prompts for bracket matching and sequence completion tasks. Your role is to enhance prompts that help solve Dyck language problems, which involve proper nesting and closure of different types of brackets ({}, <>, ()). When improving prompts, focus on these critical aspects: (1) maintaining accurate bracket pair matching, (2) preserving the LIFO (Last In First Out) order of nested structures, (3) handling multiple bracket types simultaneously, and (4) ensuring complete closure of all open brackets. You should critically analyze how the prompts can better guide the model to track open brackets, maintain proper nesting order, and systematically complete sequences. Consider incorporating pattern recognition strategies and explicit validation rules in the improved prompts. Your improvements should lead to more reliable and accurate bracket sequence completions.
> ```
>
> It can be observed that the task-specific optimizer leads to a shift in the distribution of LLM programs it tends to optimize, making it more likely to generate content that aligns with key task requirements, such as producing programs that satisfy the requirement of `preserving the LIFO (Last In First Out) order`, etc. As a result, even if the evaluator feedback is somewhat noisy or sparse, the LLM program optimized by the optimizer can still perform well, and is more likely to generate critical statements such as: `Explicitly push each opening symbol onto the stack and pop it when a corresponding closing symbol is encountered. After processing each symbol, describe the current state of the stack, focusing on unmatched opening symbols.` In contrast, if a generic optimizer is used, it becomes difficult to generate effective prompts under noisy or sparse feedback conditions.
>
> This demonstrates that adapting to a new task distribution is meaningful. **To avoid adapting to each new task distribution by hand, we meta-learn how to adapt, which is the core motivation of metaTextGrad.**
>
> Here are our responses to your questions.
>
> > 1. In the authors' response, meta-learning is beneficial when the distribution is changing. This is quite a generic statement in traditional meta-learning (before 2023). In this scenario, what does "distribution" refer to?
>
> Thank you for your question! Here, "distribution" refers to the distribution over tasks. In the traditional meta-learning formulation, we are given a distribution over tasks, and the meta-learning algorithm is supposed to train a model given a task. It is similar in our scenario. Given a specific task, we are adapting out-of-the-box generic optimizers to work well on that task. Since the optimizer needs to solve different kinds of tasks, the distribution is constantly changing, and meta-learning is beneficial when the distribution is changing.
>
> > 2. The question in weakness 2 has not been answered. Why is a task-specific optimizer necessary instead of a simpler task-specific evaluator?
>
> Thanks a lot for your question! As we pointed out in the motivation, a task-specific optimizer is necessary instead of a task-specific evaluator. This is because the feedback from a task-specific evaluator can often be sparse or noisy, making it difficult for a generic optimizer to perform effective optimization.
>
> However, if the optimizer is aligned with the tasks, then the distribution of LLM programs it optimizes will also tend to be aligned with the task requirements. In this case, the optimization becomes significantly easier, and the optimizer can still achieve good performance even when the feedback is somewhat noisy or sparse.

---

### Official Review · Reviewer_fTz6 · 2025-07-03

**Clarity:** 3
**Significance:** 2
**Originality:** 2
**Rating:** 4
**Confidence:** 4

**Summary:**

The paper proposes metaTextGrad, a meta-optimization framework designed to enhance existing language model optimizers by aligning them with specific tasks. The framework consists of two key components: a meta-prompt optimizer that refines the prompts used by LLM optimizers, and a meta-structure optimizer that improves the structural aspects of these optimizers. The method aims to systematically refine and improve the performance of LLM optimizers, with experimental results demonstrating performance improvements across various benchmarks compared to existing approaches.

**Questions:**

- What were the specific criteria for choosing the meta-optimization strategies employed in this work? Can you provide ablation studies to isolate the contributions of each component of the meta-optimizers?
- Can you provide more information throughout the inner and the outer loop optimization. E.g. performance metrics indicating consistent improvement across both - think learning curves.
- Can you provide more detailed analysis and insights into a broader range of tasks (see above)? How were parameters / prompts tuned across the different considered baselines?
- What are the potential societal impacts and limitations of this approach?

**Ethical Concerns:**

["NO or VERY MINOR ethics concerns only"]

**Final Justification:**

The authors have addressed various of my questions to my satisfaction. They added an additional experiment using ARC-AGI as well as baseline comparisons on this task. Therefore, I increased my verdict from "borderline reject" to "borderline accept".

My reasoning for the "borderline accept" verdict is threefold: First, the novelty of this work is arguably limited: Meta-optimization of optimizers is not new, but has been applied to many different settings (gradient-based/free optimization, etc.). This paper applies the approach to improve the TextGrad/Dspy optimizers. Second, the application is fairly costly, requiring multiple sequential evaluations of different LLM scaffolds on a specific task. This ultimately limits its general applicability. Finally, weaker models generally benefit the most from scaffold design, while stronger LLM models do not. Hence, it is unclear whether the introduced approach holds the test-of-time.

That being said, I believe that the work contributes to a growing body on automated agentic design and that there is a signal to be communicated to the broader NeurIPS community.

**Limitations:**

The paper acknowledges its reliance on strong instruction-following and problem analysis capabilities of advanced models, but this critical limitation requires more thorough discussion regarding its impact on the generalizability of the approach across different model architectures and capabilities. Considering more different LLM models and tasks would siginificantly strengthen the results. Additionally, the potential societal consequences of deploying such meta-optimization frameworks are not adequately addressed, which is particularly important given the broader impacts of AI technologies on various sectors and communities. The theoretical foundations of the work appear superficial and hard to transfer to real-world applications.

**Paper Formatting Concerns:**

None.

**Quality:**

3

**Strengths And Weaknesses:**

**Strengths**: The paper presents an approach to meta-optimizing language model optimizers, which directly addresses an important gap in the field. The integration of meta-prompt and meta-structure optimizers systematically improves over existing optimization methods. The comprehensive experimental evaluation across multiple benchmarks provides strong empirical evidence for the effectiveness of the proposed framework, while the hierarchical design for cost-effective resource allocation shows practical consideration for real-world deployment scenarios.

**Weaknessess**: The authors could provide more compelling justification for their choice of meta-optimization strategies, leaving readers to question why alternative approaches were not considered. The absence of detailed ablation studies prevents proper isolation of individual component contributions, making it unclear which aspects of the meta-optimizers are truly driving the performance improvements. Furthermore, I would like to see more challenging tasks considered such as ARC-AGI or coding tasks (SWE-Bench, etc.). Additionally, the discussion of potential limitations and broader impacts is insufficient, which is particularly concerning given the practical implications of deploying such optimization frameworks. Finally, the theoretical foundations appear hardly applicable to practical deployment of metaTextGrad.

---

> ### Author Rebuttal · Authors · 2025-07-29
>
> We thank the reviewer for noting that our method fills a key gap, improves over existing optimizers, is well-supported by experiments, and considers practical deployment. We will address your concerns in the following response.
>
> >  Q1. What were the specific criteria for choosing the meta-optimization strategies employed in this work?
>
> Thanks! Overall, meta-learning is particularly helpful when the distribution keeps changing. Different tasks correspond to different distributions, and what metaTextGrad essentially does is spend some compute to adapt an existing optimizer to a new distribution (e.g., aligning it with a specific task). Our criteria for choosing the meta-optimization strategies were also based on concrete experimental cases, where we observed that the optimizer's ability to align with and optimize for the task differed significantly before and after applying these meta-optimization strategies. We will support this with experiments conducted on BBH Dyck Languages. Due to space limitations, we can only present the prompt structure here; the full version will be included in the paper.
>
> **A. Optimized TGD Optimizer Demo**
>
> **TGD Optimizer prompt:**
>
> ```
> You are part of an optimization system that improves text (i.e., variable). You will be asked to creatively and critically improve prompts, solutions to problems, code, or any other text-based variable.
> ```
>
> **Meta-optimized TGD Optimizer prompt:**
>
> ```
> You are part of an optimization system specialized in improving prompts for bracket matching and sequence completion tasks. Your role is to enhance prompts that help solve Dyck language problems, which involve proper nesting and closure of different types of brackets ({}, <>, ()). When improving prompts, focus on these critical aspects: (1) maintaining accurate bracket pair matching, (2) preserving the LIFO (Last In First Out) order of nested structures,  ...
> ```
>
> **Explanation:**
>
> The TGD optimizer is generic, so its optimized program offers only general guidance like ```"ensuring that every opening bracket has a corresponding closing bracket,"``` without task-specific strategies. In contrast, the optimized TGD aligns closely with the task, emphasizing ideas like ```"preserving the LIFO (Last In First Out) order of nested structures."``` Since the optimized TGD optimizer is better aligned with the task, its optimized program also effectively implements the LIFO principle, making it more effective. This is the motivation behind our choice of this optimization strategy.
>
> **B. Optimized ADAS-TG Optimizer Demo**
>
> **ADAS-TG Optimizer prompt:**
>
> ```
> (General instructions are omitted.) Please produce only the code for the pipeline, with a more systematic or hierarchical structure if possible.
> ```
>
> **Meta-optimized ADAS-TG Optimizer prompt:**
>
> ```
> (General instructions are omitted.)  Please produce only the code for the pipeline, with the following structural improvements: 1) Implement a stack-based mechanism for tracking open brackets, 2) Create separate components for sequence parsing, bracket matching, and completion generation, 3) Include validation checks for proper nesting and bracket type matching, 4) Ensure systematic handling of different bracket types ([], {}, (), <>), ...
> ```
>
> **Explanation:**
>
> We will include the program produced by the original ADAS-TG, while the program optimized by the meta-optimized ADAS-TG can be found in Appendix pages 18–20. The original program contains generic components like a planner, reasoner, and synthesizer, lacking task-specific focus. In contrast, the optimized version adds tailored modules such as a type analyzer, stack validator, and nesting analyzer, aligning closely with BBH Dyck Languages. This is the motivation behind our choice of this optimization strategy.
>
> >  Q2. Can you provide ablation studies to isolate the contributions of each component of the meta-optimizers?
>
> In Table 6, we conducted ablation studies on BBH Dyck Languages. We conduct additional ablation experiments on BBH Word Sorting. TGD(O), ADAS-TG(O), and Struct(O) respectively denote the TGD and ADAS-TG optimizers enhanced by the meta prompt optimizer, and the optimizers enhanced by the meta structure optimizer. As shown in the table, metaTextGrad achieves the best overall performance.
>
> |Split|0-shot CoT|TGD|ADAS-TG|TGD (O)|ADAS-TG (O)|Struct (O)|metaTextGrad|
> |-|-|-|-|-|-|-|-|
> |BBH Dyck Languages Val|0.06|0.10|0.21|0.21|0.42|0.24| 0.42|
> |BBH Dyck Languages Test|0.05|0.10|0.16|0.24|0.37|0.16|0.37|
> |BBH Word Sorting Val|0.46|0.54|0.58|0.60|0.60|0.58|0.60|
> |BBH Word Sorting Test |0.55|0.55|0.58|0.59|0.55|0.61|0.65|
> |Average|0.28|0.32|0.38|0.41|0.48|0.40|0.51|
>
> >  Q3. I would like to see more challenging tasks considered.
>
> Following common practice, we use the ADAS codebase to sample ARC-AGI problems with grid sizes ≤ 5×5, forming 20 training, 30 validation, and 30 test examples. Using the official evaluation logic and initial prompt, we report one-shot success rates.
> Claude 3 Haiku serves as the program model, and Claude 3.5 Sonnet as both optimizer and meta-optimizer. As demonstrated, metaTextGrad still performs impressively. Few-shot CoT and few-shot MIPROv2 perform poorly due to model context limitations.
>
> |**Method**|**ARC-AGI**||
> |-|-|-|
> ||**Val**|**Test**|
> |**Vanilla prompting methods**||
> |Zero-shot CoT|0.27|0.23|
> |8-shot CoT|0.03|0.00|
> |Self-consistency (8)|0.30|0.23|
> |Best of N (8)|0.27|0.20|
> |**TextGrad optimizers**||
> |TGD Optimizer|0.33|0.33|
> |ADAS-TG|0.28|0.26|
> |**DSPy optimizers**||
> |Zero-shot MIPROv2|0.30|0.23|
> |8-shot MIPROv2|0.33|0.03|
> |**Meta-optimized optimizers**||
> |metaTextGrad|**0.37**|**0.40**|
>
> We provide an optimized optimizer for reference. It shows that the optimizer successfully captures key aspects of ARC-AGI, such as pattern recognition, format adherence, and structural analysis.
> Due to space limitations, we can only present the prompt structure here; the full version will be included in the paper.
>
> ```
> You are part of an optimization system specialized in improving prompts for pattern recognition and mathematical reasoning tasks. Your role is to enhance prompts that help identify and extract recurring sequences from structured data (like matrices). Focus on these key aspects when optimizing:1. Pattern Recognition: ... 2. Format Adherence: ... 3. Structural Analysis: ... 4. Edge Cases: ... 5. Comprehensive Pattern Extraction: ...
> ```
>
> >  Q4. The potential societal consequences of deploying such meta-optimization frameworks are not adequately addressed.
>
> Thanks! We agree that metaTextGrad can have meaningful societal consequences.
>
> 1. **Acceleration of domain-specific AI applications.** By making it easier to adapt LLMs to specific tasks, metaTextGrad may accelerate the deployment of more reliable AI solutions in other domains.
>
> 2. **Risk of automation bias or over-reliance.** As optimizers become more autonomous, users might rely on them without fully understanding their behavior or limitations.
>
> 3. **Potential misuse for persuasive or manipulative systems.** metaTextGrad could be exploited to generate more persuasive outputs.
>
> We will greatly expand the discussion in the paper.
>
> >  Q5. The limitation requires more thorough discussion regarding its impact on the generalizability of the approach across different model architectures and capabilities. Considering more different LLM models and tasks would siginificantly strengthen the results.
>
> Although metaTextGrad requires strong instruction-following and reasoning capabilities, many mainstream and even open-source models are already suitable. We used Qwen3-8B (non-thinking) for program execution and Qwen3-235B-A22B (non-thinking) as both optimizer and meta-optimizer on BBH Dyck Languages. metaTextGrad still significantly outperforms other baseline methods.
>
> |**Method**|**Dyck Languages**||
> |-|-|-|
> ||**Val**|**Test**|
> |**Vanilla prompting methods**||
> |Zero-shot CoT|0.27|0.27|
> |8-shot CoT|0.37|0.40|
> |Self-consistency (8)|0.31|0.32|
> |Best of N (8)|0.39|0.41|
> |**TextGrad optimizers**||
> |TGD Optimizer|0.69|0.68|
> |ADAS-TG|0.32|0.34|
> |**DSPy optimizers**||
> |Zero-shot MIPROv2|0.59|0.50|
> |8-shot MIPROv2|0.57|0.51|
> |**Meta-optimized optimizers**||
> |metaTextGrad|**0.82**|**0.77**|
>
> > Q6. The theoretical foundations appear hardly applicable to practical deployment of metaTextGrad.
>
> Thanks! Our theoretical contribution lies more in highlighting the **necessity** of performing meta-optimization. Specifically, an effectively meta-optimized optimizer can have theoretical performance guarantees, whereas an LLM optimizer without meta-optimization lacks such guarantees and may perform arbitrarily poorly.
>
> > Q7. How were parameters / prompts tuned across the different considered baselines?
>
> For all vanilla prompting methods and TextGrad optimizers, the (initial) prompts are the same as those used in metaTextGrad, which is a simple CoT prompt. For DSPy optimizers, we use the default configuration provided in ```dspy.ChainOfThought```.
>
> > Q8. Can you provide more information throughout the inner and the outer loop optimization?
>
> We conducted tests on BBH Dyck Languages and recorded the training curves for both the inner and outer loops. Since we can only describe the curves in text for the rebuttal, we provide the following summary:
>
> **Outer Loop:**
>
> 0.21, 0.42, 0.42, 0.42, 0.42, 0.42  (Yes, the result converges in the first step.)
>
> **Inner Loop:**
>
> (Outer Loop 0) 0.09, 0.09, 0.09, 0.09, 0.09, 0.14, 0.21, 0.21, 0.21, 0.21, 0.21, 0.21, 0.21, 0.21, 0.21, 0.21
>
> (Outer Loop 1) 0.07, 0.13, 0.13, 0.15, 0.15, 0.15, 0.42, 0.42, 0.42, 0.42, 0.42, 0.42, 0.42, 0.42, 0.42, 0.42
>
> To demonstrate the convergence behavior, we allocated 15 steps to the inner loop and 5 steps to the outer loop in this case. As shown, the training actually converges much earlier. Therefore, in the actual experiments, to conserve resources, we used 6 steps for the inner loop and 2 steps for the outer loop.

---

> > ### Comment · Reviewer_fTz6 · 2025-08-04
> > **Response to rebuttal**
> >
> > We thank the reviewers for their detailed answers to our questions. I appreciate the additional ARC-AGI experiment following the ADAS protocol. I will increase my score accordingly.

---

> ### Author Response · Authors · 2025-08-04
> **Thank you**
>
> We are delighted to see that you appreciate the additional ARC-AGI experiment following the ADAS protocol and will accordingly increase your score. Thank you for your effort and recognition!

---

### Author Response · Authors · 2025-08-09
**A heartfelt general comment from the authors**

Dear Reviewers and ACs,

We extend our sincere gratitude to the reviewers (fTz6, V1eb, Ss5L, rxek) for their invaluable feedback. We are pleased that the reviewers recognized **the novelty of metaTextGrad** (Reviewers Ss5L, rxek), **the importance of the research topic** (Reviewers fTz6, V1eb), **the clarity of the motivation** (Reviewers Ss5L, rxek), **the comprehensive experiments** (Reviewers fTz6, Ss5L, rxek), and **the hierarchical design for cost-effective resource allocation** (Reviewers fTz6, rxek).

**We have carefully addressed each of the reviewers’ suggestions in detail.** The main updates are as follows:

- **(Reviewers fTz6, V1eb, Ss5L)** We added insights from concrete examples to help readers understand what happens after meta-optimization, what benefits can be gained when adapting to a specific task, and why a task-specific prompt optimizer is necessary. We also explained the specific criteria for selecting the meta-optimization strategies used in this work.

- **(Reviewers fTz6, Ss5L, rxek)** We recorded the training curves for both the inner and outer loops of metaTextGrad to facilitate understanding of its convergence characteristics.

- **(Reviewers fTz6, V1eb)** We included experiments using the Qwen model as the optimizer, demonstrating the effectiveness and generalizability of metaTextGrad across different models.

- **(Reviewers V1eb, Ss5L)** We clarified that our benchmark selection follows the choices made in prior published works.

- **(Reviewer fTz6)** We conducted more detailed ablation studies to isolate the contributions of each component of the meta-optimizers.

- **(Reviewer fTz6)** We included experiments on the more challenging ARC-AGI task, showing that metaTextGrad remains effective even in difficult scenarios.

- **(Reviewer Ss5L)** We provided a detailed cost analysis of metaTextGrad compared with each baseline.

We sincerely appreciate the reviewers’ and ACs' time and effort. All discussions and experimental results have been incorporated into the final version of the paper. During the rebuttal period, we have made every effort and demonstrated our utmost sincerity in addressing the reviewers’ concerns. Once again, we deeply appreciate the reviewers’ and ACs' time and insightful feedback.

With best regards,

The Authors

---

### Note · Authors · 2025-08-13

Dear Reviewers, AC, SAC, and PC,

We would like to express our sincere gratitude to all four reviewers for their invaluable participation in the rebuttal process and for providing constructive feedback on our work. We also extend our deep appreciation to the AC for their continuous engagement and support throughout this process.

During the rebuttal period, **we conducted a series of additional experiments to the best of our ability, which have been positively acknowledged by the reviewers**.

We are pleased to learn that we have resolved all the concerns of reviewers fTz6 and Ss5L.

> Reviewer  fTz6 commented: "I appreciate the additional ARC-AGI experiment following the ADAS protocol. I will increase my score accordingly."

> Reviewer   Ss5L commented: "Thanks for explanation. I do not have other concerns and I suggest all of these experimental results be incorporated in the next version, including cost analysis and in-depth case study, etc. I will raise my overall rating to 5."

Though reviewer rxek has not yet submitted the final rating, we are glad to know that we have resolved all their concerns and that they will further increase the score.

> Reviewer   rxek commented: "Dear authors, thank you for addressing my concerns and questions. I have increased my score."

We are pleased to see that we have addressed some of reviewer V1eb’s concerns in the first-round rebuttal. Although we did not receive feedback from the second-round discussion, in the second round we provided a detailed explanation of our work’s motivation, why meta-learning is beneficial, and why a task-specific optimizer is necessary instead of a simpler task-specific evaluator. We sincerely hope that the reviewers and the AC are satisfied with this explanation.

> Reviewer V1eb commented: "Thanks for the detailed rebuttal. It addressed some of my concerns, ... My final score on this paper will depend on the authors' response."

**This has been an incredibly rewarding and meaningful rebuttal process.** Throughout **more than 30** conversations with the reviewers, we carefully **addressed each concern raised**. We are truly grateful to the SAC, AC, and reviewers for dedicating your valuable time and effort.

Best regards,

The authors

---

### Decision · Program_Chairs · 2025-09-17

**Decision:**

Accept (poster)

**Comment:**

This paper introduces metaTextGrad, a meta-optimization framework designed to improve existing LLM optimizers by aligning them more closely with specific tasks. The framework includes two main components: (1) a meta-prompt optimizer that refines prompts used by LLM optimizers, and (2) a meta-structure optimizer that improves the combination.

Strengths:

1. The integration of meta-prompt and meta-structure optimizers is novel and effective, enabling both prompt-level refinement and structural optimization.

2. Empirical results are strong: consistent gains over multiple datasets and well-chosen baselines and analysis of cost efficiency and resource usage.

Weaknesses (some has been addressed in the rebuttal, and it will be better to add to the main paper):

1. Ablation and analysis are insufficient: the contributions of each component (meta-prompt vs. meta-structure) are not isolated, making it unclear which is the primary driver of improvement.

2. Benchmark coverage is somewhat limited: only a subset of BBH tasks are tested, and more challenging domains (e.g., ARC-AGI, SWE-Bench) could strengthen the claims.

Overall, the reviewers agree that this paper contributes to auto prompt design. I recommend acceptance.